# Revisiting Adversarial Robustness Distillation from the Perspective of Robust Fairness

**Xinli Yue, Ningping Mou, Qian Wang, Lingchen Zhao**[*]
Key Laboratory of Aerospace Information Security and Trusted Computing, Ministry of
Education, School of Cyber Science and Engineering, Wuhan University, Wuhan 430072, China
{xinliyue, ningpingmou, qianwang, lczhaocs}@whu.edu.cn

## Abstract

Adversarial Robustness Distillation (ARD) aims to transfer the robustness of large
teacher models to small student models, facilitating the attainment of robust perfor-
mance on resource-limited devices. However, existing research on ARD primarily
focuses on the overall robustness of student models, overlooking the crucial aspect
of *robust fairness*. Specifically, these models may demonstrate strong robustness
on some classes of data while exhibiting high vulnerability on other classes. Un-
fortunately, the "buckets effect" implies that the robustness of the deployed model
depends on the classes with the lowest level of robustness. In this paper, we first
investigate the inheritance of robust fairness during ARD and reveal that student
models only partially inherit robust fairness from teacher models. We further vali-
date this issue through fine-grained experiments with various model capacities and
find that it may arise due to the gap in capacity between teacher and student models,
as well as the existing methods treating each class equally during distillation. Based
on these observations, we propose **Fair A**dversarial **R**obustness **D**istillation (Fair-
ARD), a novel framework for enhancing the robust fairness of student models by
increasing the weights of difficult classes, and design a geometric perspective-based
method to quantify the difficulty of different classes for determining the weights.
Extensive experiments show that Fair-ARD surpasses both state-of-the-art ARD
methods and existing robust fairness algorithms in terms of robust fairness (e.g., the
worst-class robustness under AutoAttack is improved by at most 12.3% and 5.3%
using ResNet18 on CIFAR10, respectively), while also slightly improving overall
robustness. Our code is available at: https://github.com/NISP-official/Fair-ARD.

## 1 Introduction

Knowledge distillation [15, 29, 19, 5] is currently a popular technique for transferring knowledge
from large models (teacher models) to small models (student models). This practice is particularly
vital in resource-constrained environments such as mobile or embedded devices. However, as
neural networks are notably vulnerable to adversarial examples [31, 10], where the addition of small
perturbations to input examples can potentially lead to misclassification, student models obtained
through knowledge distillation are also susceptible to these adversarial attacks [4, 7, 2, 8]. Therefore,
Adversarial Robustness Distillation (ARD) [9, 44, 45, 43] have been proposed to train robust student
models by integrating adversarial training (AT) [24, 40, 34, 37].

However, some recent works [38, 33] have revealed the issue of *robust fairness* in AT, where
adversarially trained models may exhibit high robustness on some classes while demonstrating
significantly low robustness on others. Due to the "buckets effect," the security of a system often
depends on the security of its weakest component. Therefore, even if the overall robustness of a

---

[*]Corresponding author.

model is high, poor robustness on a specific class of data can pose security issues. For instance, an autonomous driving system that demonstrates high robustness for inanimate objects on the road but lacks robustness when detecting pedestrians may be misled by adversarial examples, leading to traffic accidents.

Thus, we raise a question: *how about the robust fairness of ARD that also incorporates adversarial examples into training like AT?* Previous works have only concentrated on the overall robustness of ARD, overlooking the aspect of robust fairness. In this paper, we, for the first time, investigate whether ARD can transfer robust fairness from teacher models to student models. Unfortunately, experimental results provide a negative answer to this question, revealing that student models can only partially inherit robust fairness from their teacher counterparts.

Then, we reveal that the main reason lies in the capacity gap between the teacher model and the student model, and the common strategy of the prior works that treats the knowledge about each class in the same way during the distillation process. For instance, in the CIFAR-10 dataset [20], the "car" class is relatively easy for the teacher model to learn, and the student model can fit it well. By contrast, for the "cat" class, learning its knowledge is already challenging for the teacher model. Therefore, expecting a student model with significantly reduced capacity to attain the same level of robustness as the teacher model becomes even more difficult. This implies that the equitable treatment of each class in previous methods [9, 44, 45, 43] may be inappropriate regarding robust fairness. It becomes challenging for the student model to effectively learn the knowledge from the teacher model regarding difficult classes.

In light of these observations, we argue for treating different classes distinctively during distillation, focusing more on difficult classes and less on easy ones (those student models can fit relatively easily). While previous methods [38] have incorporated re-weighting schemes to address robust fairness in AT, a closer examination of their quantification of example difficulty suggests that their metric for measuring class difficulty is intuitively coarse-grained, potentially lacking precision in assessing the actual difficulty of classes. From a geometric perspective, examples farther from the decision boundary are less likely to be misclassified, thereby implying lower difficulty. Therefore, the distance from an example to the decision boundary can serve as a metric for example difficulty. Comparative analysis indicates that this metric provides finer granularity. Additionally, the average difficulty of examples within a class can be employed to measure the class's difficulty. Drawing inspiration from these insights, we introduce a novel, more fine-grained class difficulty metric. We further devise a unique re-weighting scheme, adaptively assigning different weights to classes with varying levels of difficulty during distillation, thereby mitigating the bias of student models towards knowledge of different classes.

Our contributions can be summarized as follows:

- To the best of our knowledge, we are the first to investigate the issue of robust fairness in adversarial robustness distillation. We reveal that the student model only partially inherits the robust fairness of the teacher model and present the internal reasons.

- We present the Fair Adversarial Robustness Distillation (Fair-ARD) framework, which employs a more refined difficulty metric and an adaptive class re-weighting approach, enabling the student model to learn the knowledge about each class from the teacher model in a fairer way. Fair-ARD is a versatile method: existing ARD techniques such as IAD [44], RSLAD [45], and MTARD [43] can also be modified into Fair-IAD, Fair-RSLAD, and Fair-MTARD, respectively.

- Extensive experiments validate that Fair-ARD can significantly improve the robust fairness of the student model and slightly improve the overall robustness, outperforming state-of-the-art ARD methods and existing robust fairness algorithms.

## 2 Can the Student Model Inherit the Robust Fairness of the Teacher Model?

In this section, we investigate the capability of the student model to inherit robust fairness from the teacher model. Due to the space limitation, we extensively introduce the related works in Appendix B, including AT, ARD, robust fairness, etc.

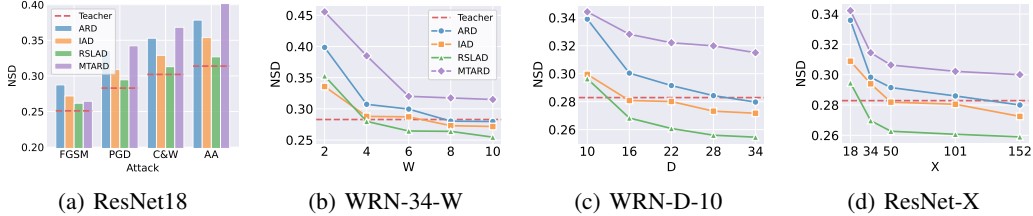

|          |          |          |          |
|:--------:|:--------:|:--------:|:--------:|
| (a) ResNet18 | (b) WRN-34-W | (c) WRN-D-10 | (d) ResNet-X |

Figure 1: The NSD of teacher and student models on CIFAR10. (a) The NSD of the teacher model WideResNet-34-10 [39] and student model ResNet18 [13] generated by four ARD methods under four adversarial attacks. (b)~(d) The NSD of teacher (WideResNet-34-10) and student models with different capacities generated by four ARD methods under PGD-20 attack [24], where (b) the horizontal axis W represents the widen factor of WideResNet; (c) the horizontal axis D represents the depth of WideResNet; (d) the horizontal axis X represents the number of layers in ResNet.

Table 1: The percentage of the robustness for different classes inherited by the student model ResNet-18 with respect to the corresponding class robustness of the teacher model WideResNet-34-10.

| Method | FGSM | | PGD | | C&W | | AA | |
|:------:|:------:|:------:|:------:|:------:|:------:|:------:|:------:|:------:|
| | cat | car | cat | car | cat | car | cat | car |
| ARD | 85.15% | **94.95%** | 80.60% | **92.78%** | 80.74% | **93.59%** | 72.53% | **93.13%** |
| IAD | 84.55% | **96.39%** | 81.72% | **94.42%** | 86.89% | **95.26%** | 79.83% | **94.69%** |
| RSLAD | 90.30% | **95.07%** | 88.81% | **93.79%** | 92.62% | **94.36%** | 89.70% | **94.56%** |
| MTARD | 89.70% | **99.28%** | 75.37% | **96.70%** | 71.31% | **96.41%** | 63.52% | **96.11%** |

## 2.1 The Degree of Robust Fairness Inheritance

Prior works about ARD [9, 44, 45, 43] mainly focused on improving the overall robustness of the student model. However, following the observation of robust fairness concerns with AT, the usability of ARD should be carefully evaluated to avoid potential security risks. In this section, we present a detailed analysis of the ARD methods in robust fairness.

Specifically, we evaluate the Normalized Standard Deviation (NSD[2]) of the class-wise robustness for the teacher model and the student models trained by four ARD methods: ARD [9], IAD [44], RSLAD [45], and MTARD [43]. The results are shown in Fig. 1(a). The details about hyperparameters and the methods for generating adversarial examples are in Section 4.1. The results show that all the student models consistently achieve higher NSD than the teacher model, i.e., the robust fairness of the student models is worse than the teacher model. In other words, student models can only partially inherit the teacher model's robust fairness.

## 2.2 Influential Factors in Robust Fairness Inheritance

It is well-known that the gap in model capacity accounts for the inferior performance of the student model compared to the teacher model in knowledge distillation [25, 5]. Naturally, this leads us to consider whether the robust fairness inheritance is also related to this gap. We vary the capacity of the student model (by adjusting the depth or width of the model [28]) and analyze the change of NSD. As shown in Fig. 1 (b) to (d), for a given teacher model, the NSD decreases with the increasing of the student model's capacity, approaching or even falling below the teacher model's NSD. This confirms our hypothesis that the capacity gap between models leads to the inferior robust fairness of the student model compared to the teacher model.

Next, we conduct a more detailed evaluation of the relationship between the capability of the student model and the robust fairness it inherits. We calculate the proportion of robustness inherited by the student model for each class from the teacher model. Table 1 presents the results for the least robust class (cat) and the most robust class (car) in the teacher model. We can observe that the hardest class

---

[2]NSD is a metric proposed in [38] to measure the robust fairness of a model; the larger NSD, the worse the robust fairness. $NSD = SD/mean$, where SD is the Standard Deviation of class-wise robustness.

(cat) inherits less robustness than the easiest class (car). It reveals that the learning difficulty for the student model varies among different classes. For instance, the "car" class, whose knowledge is relatively easy, is well fitted by the student model with smaller capacity, inheriting about 94% of the teacher model's robustness under AutoAttack (AA) [8] for these four distillation methods. In contrast, the "cat" class presents increased difficulty, with the teacher model only achieving 26.8% robustness under PGD-20 [24]. This decline amplifies the limitation for the student model, which only inherits less than 80% of the teacher's robustness under AA for ARD, IAD, and MTARD. More results about other classes are available in Appendix C.2. Therefore, considering the inherent difference in difficulty among different classes, the practice of prior ARD methods that treat all classes equally poses an obstacle to the inheritance of robust fairness. This could potentially result in a case where student models may effectively learn knowledge of easy classes while hard to learn knowledge of difficult classes. Consequently, we should assign larger (smaller) weights to harder (easier) classes, allowing student models to focus more on the teacher model's knowledge about hard classes.

## 3 Methodology: Fair Adversarial Robustness Distillation

In Section 2, we observe that treating all classes equally poses an obstacle to the inheritance of robust fairness. An intuitive way to mitigate this issue is to assign different weights to different classes. Let $\omega_i$ be the weight for the $i$-th class; we formulate a fair adversarial robustness distillation (Fair-ARD) method as follows:

$$\min_{\theta_S} \frac{1}{C} \sum_{i=1}^{C} \frac{1}{n_i} \sum_{j=1}^{n_i} \omega_i \mathcal{L}_{\mathrm{ARD}} \left( S, T, x_i^j, y_i, \tau, \alpha \right), \tag{1}$$

$$\mathcal{L}_{\mathrm{ARD}} \left( S, T, x_i^j, y_i, \tau, \alpha \right) = (1 - \alpha) \, \mathrm{CE} \left( S^\tau \left( x_i^j \right), y_i \right) + \alpha \tau^2 \mathrm{KL} \left( S^\tau \left( \tilde{x}_i^j \right), T^\tau \left( x_i^j \right) \right), \tag{2}$$

where $x_i^j$ is the $j$-th example of the $i$-th class, $\tilde{x}_i^j$ is the adversarial example of $x_i^j$, $y_i$ is the label of the $i$-th class, $n_i$ is the number of examples of the $i$-th class, $C$ is the total number of classes, $\mathcal{L}_{\mathrm{ARD}}$ is the loss of ARD [9], $S$ is the student model with parameter $\theta_S$, $T$ is the teacher model, $\tau$ is the temperature constant added to the softmax operation, $\alpha$ is a hyperparameter in ARD, CE is the Cross-Entropy loss, and KL is the Kullback-Leibler divergence loss.

### 3.1 Metrics to Measure Class Difficulty

The main challenge is determining the optimal weights. Hence, it is necessary to propose a metric for quantifying class difficulty to calculate the weights. Prior works about improving the robust fairness of AT, such as Fair Robust Learning (FRL) [38], measure the class difficulty by the class-wise robustness, i.e., the expectation of the successful defense of adversarial examples within a given class. Let $r_i$ represent the robustness of the $i$-th class, and it can be formulated as

$$r_i = \frac{1}{n_i} \sum_{j=1}^{n_i} \varphi \left( x_i^j, y_i \right), \tag{3}$$

$$\varphi(x, y) = \mathbb{1} \left( S \left( \tilde{x} \right) = y \right), \tag{4}$$

where $\tilde{x}$ is the adversarial example of the natural example $x$, $y$ is the label of $x$. $\mathbb{1} \left( \cdot \right)$ is the indicator function, which outputs 1 when $S \left( \tilde{x} \right) = y$ and 0 otherwise. Thus, the difficulty of an example, denoted by $\varphi \left( x, y \right)$, can only be 1 or 0, representing defense success or failure, respectively. And we refer to this metric of example difficulty as FRL Example Difficulty (**FED**).

However, we are concerned that such a coarse-grained method of quantifying example difficulty can be further refined to obtain more precise results. In Support Vector Machines (SVMs) [6], the distance of an example to the decision boundary can be utilized to measure the difficulty of that example, offering a broader range and finer granularity in difficulty measurement. For linear SVMs, where the decision boundary is a hyperplane, distance computation is relatively straightforward. However, in DNNs, the decision boundary in high-dimensional space might constitute a complex, non-linear surface, rendering the calculation of example-to-boundary distance challenging. Inspired by GAIRAT [41], the least PGD steps (**LPS**), representing the steps needed for adversarial perturbations of a natural example to cross the decision boundary, can be employed as an approximation for the

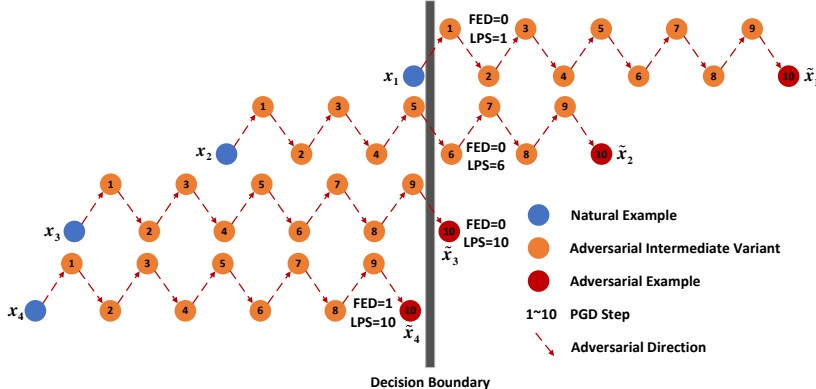

Figure 2: Comparison with the example difficulty metrics FED, adopted by FRL [38] and LPS [41]. $x_1$, $x_2$, $x_3$, and $x_4$ are natural examples from the same class. Adversarial examples of $x_1$, $x_2$, and $x_3$ can cross the decision boundary, indicating defense failure. Thus, their FED values are all 0. However, the required PGD steps for successful attacks differ for $x_1$, $x_2$, and $x_3$, resulting in their LPS values being 1, 6, and 10, respectively. The adversarial example of $x_4$ does not cross the decision boundary, indicating a successful defense. This results in a FED value of 1 and an LPS value of 10 for $x_4$.

distance from the example to the decision boundary. Specifically, for data point $(x, y)$, the distance $d(x, y)$ from the data point $(x, y)$ to the decision boundary is calculated as follows:

$$\begin{cases} \tilde{x}^{(t+1)} = \Pi_{\mathcal{B}_\epsilon[x]} \left( \tilde{x}^{(t)} + \gamma \operatorname{sign} \left( \nabla_{\tilde{x}^{(t)}} \ell \left( S \left( \tilde{x}^{(t)} \right), y \right) \right) \right) \\ d(x, y) = \underset{t \in [0, K]}{\operatorname{argmin}} \left( S \left( \tilde{x}^{(t)} \right) \neq y \right), \end{cases} \tag{5}$$

where $\tilde{x}^{(t)}$ is the adversarial example of the natural example $x$ at PGD step $t$, $K$ is the number of steps, $\gamma$ is the step size, $\ell$ is the loss function, e.g. CE loss, and $\Pi_{\mathcal{B}_\epsilon[x]}(\cdot)$ is the projection function that projects the adversarial data back into the $\epsilon$-ball centered at $x$ when necessary.

Fig. 2 presents an example to show the difference between FED and LPS. For three different natural examples, $x_1$, $x_2$, and $x_3$, their FED values are all 0, but their LPS values are 1, 6, and 10, respectively. Thus, LPS can effectively distinguish difficult levels between different examples, providing a more fine-grained measure for quantifying example difficulty than FED. We will provide more experimental results to show the differences between the two metrics in Section 5.

Then, the average LPS of all examples within a class can serve as a metric of class difficulty. Therefore, let $\kappa_i$ denote the average LPS for all examples in the $i$-th class, and it can be calculated as follows:

$$\kappa_i = \frac{1}{n_i} \sum_{j=1}^{n_i} d \left( x_i^j, y_i \right). \tag{6}$$

## 3.2 The Re-weighting Strategy

After defining the metric for measuring class difficulty, we need to design a re-weighting function to allocate appropriate weights for each class based on its difficulty. Following the principle of giving larger (smaller) weights to harder (easier) classes, the weight $\omega_i$ should decrease w.r.t. $\kappa_i$. Therefore, we heuristically design several re-weighting functions, and here we provide an example. More examples of re-weighting functions are discussed in Appendix C.8.

$$\omega_i = \frac{1}{\kappa_i^\beta}, \tag{7}$$

where $\beta$ is a hyperparameter controlling the smoothness of the re-weighting function. To make the total loss roughly in the same scale when applying $\omega_i$, we normalize $\omega_i$ so that $\frac{1}{C} \sum_{i=1}^{C} \omega_i = 1$.

Formally, given an example from the $i$-th class, we propose to add a weight factor of $1/\kappa_i^\beta$ in the loss function, where the hyperparameter $\beta \in [0, \infty)$. If $\beta = 0$, then $\omega_i = 1$, corresponding to no re-weighting, i.e., Fair-ARD recovers to vanilla ARD with equal weights for all classes.

Table 2: The average (Avg.) and worst-class (Worst) robustness for various algorithms using ResNet18 on CIFAR-10. Better results in comparison with vanilla ARD and our proposed fair version are **bolded**.

| Method | Clean | | FGSM | | PGD | | C&W | | AA | |
|---|---|---|---|---|---|---|---|---|---|---|
| | Avg. | Worst | Avg. | Worst | Avg. | Worst | Avg. | Worst | Avg. | Worst |
| Natural | 94.35 | 87.40 | 16.13 | 5.30 | 0.00 | 0.00 | 0.00 | 0.00 | 0.00 | 0.00 |
| SAT | 84.27 | 64.10 | 56.81 | 28.30 | 49.11 | 22.10 | 48.58 | 20.50 | 46.13 | 17.40 |
| TRADES | 82.22 | 64.80 | 58.38 | 31.00 | 52.35 | 25.80 | 50.33 | 22.30 | 49.01 | 21.20 |
| ARD | **83.22** | 61.00 | **58.77** | 28.10 | 51.65 | 21.60 | **51.25** | 19.70 | 49.05 | 16.90 |
| Fair-ARD (ours) | 82.96 | **68.10** | 57.69 | **39.20** | **52.05** | **33.20** | 50.69 | **31.00** | **49.13** | **29.20** |
| IAD | **83.25** | 60.60 | **58.90** | 27.90 | 52.08 | 21.90 | 51.01 | 21.20 | 48.95 | 18.60 |
| Fair-IAD (ours) | 83.19 | **68.70** | 58.31 | **36.50** | **52.27** | **30.00** | **51.16** | **28.10** | **49.28** | **25.60** |
| RSLAD | 83.04 | 62.70 | 60.03 | 29.80 | 54.13 | 23.80 | 52.76 | 22.60 | 51.18 | 20.90 |
| Fair-RSLAD (ours) | **83.59** | **67.80** | **60.16** | **35.50** | **54.33** | **29.70** | **53.07** | **26.90** | **51.23** | **25.10** |
| MTARD | **87.14** | **69.80** | **60.62** | 30.10 | 50.81 | 20.70 | 48.85 | 18.00 | 46.10 | 16.10 |
| Fair-MTARD (ours) | 81.98 | 62.10 | 59.11 | **30.60** | **53.96** | **27.50** | **52.32** | **24.20** | **50.60** | **22.60** |

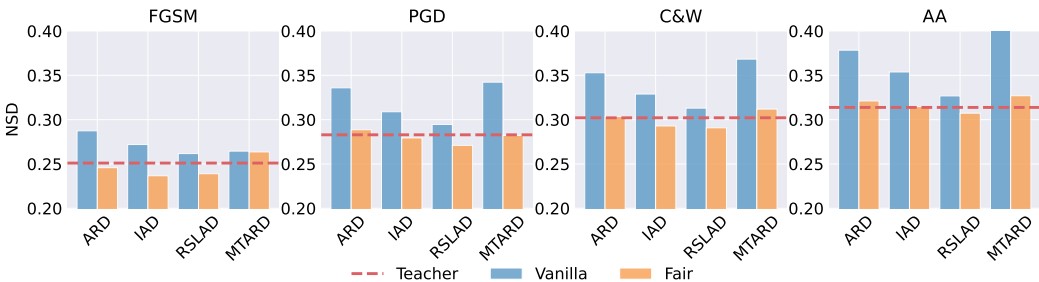

Figure 3: The NSD of the vanilla ARD (Vanilla) and our proposed fair version (Fair) using ResNet18 on CIFAR-10. The red line represents the NSD of the teacher model (WideResNet34-10). From left to right, the panels show results under FGSM, PGD, C&W, and AA, respectively.

Due to the requirement of computing the weights for each class based on the entire dataset, $\omega_i$ cannot be updated by mini-batch like GAIRAT. Instead, it is calculated based on the model from the previous epoch. To demonstrate the generality of the proposed re-weighting strategy, we apply it to four ARD methods: ARD [9], IAD [44], RSLAD [45], and MTARD [43]. And the algorithms of their fair version Fair-ARD, Fair-IAD, Fair-RSLAD, and Fair-MTARD are in Appendix A.

## 4 Experiments

### 4.1 Experimental Setup

**Baselines.** We conduct the experiments on four benchmark datasets, CIFAR-10, CIFAR-100 [20], SVHN [27], and Tiny-ImageNet [21]. Due to the space limitation, results for datasets beyond CIFAR-10 are presented in Appendix C. We consider seven baseline methods, including natural training, two AT methods: SAT [24] and TRADES [40], and four distillation methods: ARD[9], IAD[44], RSLAD[45], and MTARD[43].

**Teacher-Student Pairs.** Following [45, 43], the used teacher models include a naturally trained ResNet-56 [13] and a robust WideResNet34-10 [39] trained with TRADES. Student models include ResNet-18 and MobileNetV2 [30].

**Training Settings.** We use the stochastic gradient descent (SGD) optimizer with an initial learning rate of 0.1, a momentum of 0.9, and a weight decay of 2e-4 to train the networks. The batch size is set as 128. For the baseline methods, i.e., SAT, TRADES, ARD, IAD, RSLAD, and MTARD, we

Table 3: The average (Avg.) and worst-class (Worst) robustness for various algorithms using MobileNetV2 on CIFAR-10. Better results in comparison with vanilla ARD and our proposed fair version are **bolded**.

| Method | Clean | | FGSM | | PGD | | C&W | | AA | |
|---|---|---|---|---|---|---|---|---|---|---|
| | Avg. | Worst | Avg. | Worst | Avg. | Worst | Avg. | Worst | Avg. | Worst |
| Natural | 92.22 | 82.40 | 9.26 | 0.40 | 0.00 | 0.00 | 0.00 | 0.00 | 0.00 | 0.00 |
| SAT | 81.36 | 55.50 | 55.88 | 24.10 | 49.97 | 18.10 | 48.70 | 16.50 | 46.10 | 12.40 |
| TRADES | 80.44 | 63.20 | 56.40 | 30.20 | 51.14 | 25.80 | 48.38 | 21.30 | 47.39 | 20.50 |
| ARD | 81.15 | 59.20 | 56.44 | 26.40 | 50.74 | 22.00 | **49.96** | 21.10 | **48.20** | 18.60 |
| Fair-ARD (ours) | **82.09** | **69.40** | **56.48** | **36.70** | **50.94** | **29.80** | 49.69 | **26.40** | 48.03 | **24.00** |
| IAD | 81.07 | 56.80 | **56.73** | 26.30 | 51.57 | 22.50 | **49.75** | 19.90 | **48.02** | 17.40 |
| Fair-IAD (ours) | **81.48** | **66.90** | 56.72 | **38.60** | **51.75** | **32.60** | 49.42 | **28.90** | 47.72 | **27.10** |
| RSLAD | 82.97 | 62.60 | 58.67 | 29.30 | 52.91 | 24.10 | 51.58 | 21.50 | 49.93 | 19.60 |
| Fair-RSLAD (ours) | **83.56** | **67.10** | **59.14** | **34.90** | **53.50** | **30.20** | **52.15** | **28.10** | **50.22** | **26.30** |
| MTARD | **87.78** | **71.40** | 56.09 | 24.60 | 43.15 | 13.20 | 42.17 | 12.30 | 39.42 | 9.80 |
| Fair-MTARD (ours) | 81.97 | 63.00 | **58.20** | **31.40** | **53.26** | **27.90** | **51.16** | **24.90** | **49.66** | **22.80** |

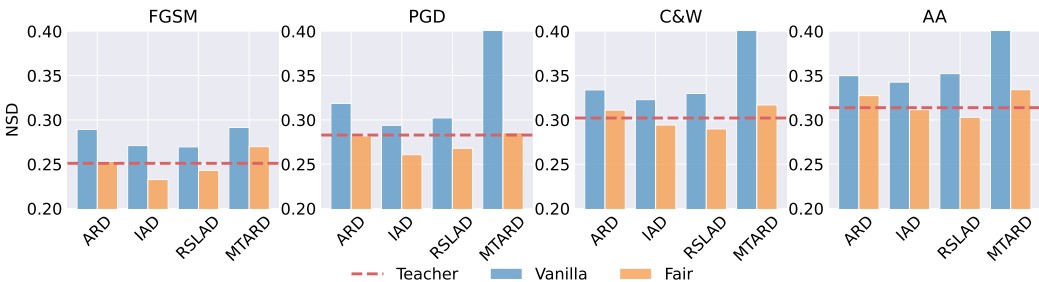

Figure 4: The NSD of the vanilla ARD (Vanilla) and our proposed fair version (Fair) using MobileNetV2 on CIFAR-10. The red line represents the NSD of the teacher model (WideResNet34-10). From left to right, the panels show results under FGSM, PGD, C&W, and AA, respectively.

strictly follow their original settings. For the version improved by our proposed method, namely Fair-ARD, Fair-IAD, Fair-RSLAD, and Fair-MTARD, we also follow the original settings of the non-fair training version. Additionally, we search for the hyperparameter $\beta$ on CIFAR10 using Fair-ARD and determine $\beta = 2$. We adopt $\beta = 2$ for all other fair adversarial robustness distillation methods and other datasets. All the results are reported as the average of five runs, and standard deviations are omitted because they are too small.

**Evaluation Metrics.** We evaluate the model using four typical attacks: FGSM [10], PGD [24], C&W [4], and AutoAttack (AA) [8]. For both datasets, the maximum perturbation is set to $\epsilon = 8/255$, and the perturbation steps for both PGD and C&W are set to 20 following [45]. And following [38], we use the worst-class robustness and NSD to measure the robust fairness of the student model. The average and worst-class robustness of teacher models under the four attacks can be found in Appendix C.1.

## 4.2 Robustness and Fairness

As shown in Table 2 and Table 3, after applying Fair-ARD, the models have a significant improvement in robust fairness. In addition, in most of the results, there is a slight improvement in the average accuracy and robustness as well. Our method achieves state-of-the-art worst-class robustness under all attacks, surpassing even the worst-class robustness of the teacher model. These results confirm that Fair-ARD outperforms all baseline methods in inheriting robust fairness from the teacher model. Considering that we fine-tune the hyperparameter using ResNet-18 on Fair-ARD, it demonstrates the most significant enhancement, with the worst-class robustness under AA experiencing a substantial

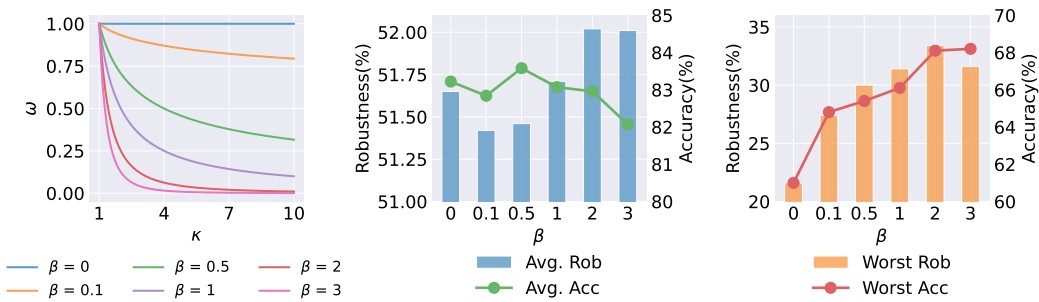

Figure 5: Effects of different $\beta$ on Fair-ARD using ResNet-18 on CIFAR10. *Left*: $\omega$ values under different $\beta$. *Middle*: the average robustness and accuracy of student models using different $\beta$. *Right*: the worst-class robustness and accuracy of student models using different $\beta$.

12.3% improvement. If the hyperparameter were fine-tuned on other fair methods, e.g., Fair-IAD, they could potentially exhibit further improvements.

For MTARD, we observe that while Fair-ARD can improve its robust fairness, it may decrease clean accuracy. This is due to the workflow of MTARD. The student model is simultaneously guided by a clean model and a teacher model, where the weights of these two models sum up to 1. And Fair-ARD aims to enhance robust fairness. Consequently, the robust teacher's weight is potentially increased during the training process, naturally leading to a decrease in the clean teacher's weight, thereby reducing the clean accuracy.

Then, we present the NSD of both the vanilla ARD and Fair-ARD across various models in Fig. 3 and Fig. 4. The results show that Fair-ARD effectively reduces the NSD of the student model, especially in most experimental results where the NSD of the student model even falls below that of the teacher model. This further demonstrates the superiority of our method in enhancing robust fairness.

### 4.3 Ablation Studies

To obtain a comprehensive understanding of Fair-ARD, we investigate the effect of $\beta$. Additional ablation studies on different re-weighting functions and teacher models are presented in Appendix C.8 and C.9, respectively. Specifically, we choose ResNet-18 as the student model and ARD as the baseline method; other settings are the same as in Section 4.1. We first visualize the value of $\omega$ with different $\beta$ in the left panel of Fig. 5. With the increase of $\beta$, the re-weighting magnitude and the weight gap among classes also increase, and student models will pay more attention to the knowledge of difficult classes.

In the middle and right panels of Fig. 5, we present the average and worst-class performance of the student model obtained with different $\beta$, respectively. Note that when $\beta = 0$, all classes are considered equally, i.e., Fair-ARD is the same as ARD in this case. We can observe that the average robustness of Fair-ARD is worse than ARD when $\beta$ is too small. However, when $\beta > 1$, Fair-ARD exhibits higher robustness compared to ARD while almost not sacrificing accuracy. Additionally, the worst-class robustness and accuracy of Fair-ARD across various $\beta$ values surpass those of ARD, highlighting the superiority of our method. In particular, when $\beta = 2$, the student model achieves the highest worst-class robustness. Therefore, we adopt this setting in our experiments. Moreover, the worst-class robustness of Fair-ARD initially increases and then decreases as $\beta$ increases. The reason might be that an excessively large $\beta$ leads to a significant weight disparity among classes, resulting in unstable training.

## 5 Discussion

In this section, we discuss the differences between our Fair-ARD and related works about (fair) adversarial training with detailed experimental results. More discussions are provided in Appendix C.10.

Table 4: The average (Avg.) and worst-class (Worst) robustness for various algorithms using ResNet-18 on CIFAR-10. The best results are **bolded**, and the second best results are underlined.

| Method | Clean | | FGSM | | PGD | | C&W | | AA | |
|---|---|---|---|---|---|---|---|---|---|---|
| | Avg. | Worst | Avg. | Worst | Avg. | Worst | Avg. | Worst | Avg. | Worst |
| ARD | **83.22** | 61.00 | **58.77** | 28.10 | 51.65 | 21.60 | **51.25** | 19.70 | 49.05 | 16.90 |
| GAIR-ARD | 81.09 | 65.80 | 54.04 | 34.60 | 50.00 | 32.60 | 42.69 | 20.70 | 40.88 | 18.90 |
| FRL-ARD | 81.47 | 61.40 | 56.90 | 34.30 | 50.24 | 28.60 | 49.76 | 26.80 | 47.84 | 23.90 |
| FAT-ARD | 82.98 | 65.20 | 57.62 | 38.00 | 51.11 | 31.00 | 49.99 | 29.20 | 48.22 | 25.90 |
| CFA-ARD | 82.00 | 62.30 | 57.68 | 30.60 | **52.43** | 25.00 | 51.23 | 22.30 | **49.73** | 20.00 |
| Fair-ARD (ours) | 82.96 | **68.10** | 57.69 | **39.20** | 52.05 | **33.20** | 50.69 | **31.00** | 49.13 | **29.20** |

## 5.1 Comparison with GAIRAT

GAIRAT [41] is an instance-reweighted adversarial training algorithm, and our metric for measuring class difficulty is designed based on the LPS proposed by GAIRAT. We highlight several differences between our Fair-ARD and GAIRAT. Firstly, Fair-ARD focuses on both overall robustness and robust fairness, while GAIRAT only concerns overall robustness. Additionally, GAIRAT can be viewed as finding solutions for each individual example, whereas Fair-ARD seeks solutions for an entire class of examples. Finally, the re-weighting functions of the two are also different: GAIRAT' function is a tanh-type function, while ours is a power function. And the comparison of different re-weighting functions is provided in Appendix C.8.

For a fair comparison, we applied the GAIRAT method to ARD, resulting in a variant we refer to as GAIR-ARD. As shown in Table 4, although GAIR-ARD demonstrates higher worst-class robustness under the PGD attack than ARD, it still does not outperform our Fair-ARD. Furthermore, under the AutoAttack, the average robustness and robust fairness of GAIR-ARD are both significantly lower than our Fair-ARD. Upon investigating the reason, we observed the weights assigned to each example during the training process of GAIR-ARD and found that the disparities among different examples' weights were substantial, even differing by several orders of magnitude. This may lead to subtle gradient masking [3, 16], resulting in a substantial drop in robustness when facing AA. Instead, our Fair-ARD employs a re-weighting function that acts on the classes, and the hyperparameter $\beta$ can control the re-weighting amplitude within a smaller range. Consequently, Fair-ARD does not suffer from gradient masking, as evidenced by the results under AA.

## 5.2 Comparison with Robust Fairness Algorithms

FRL [38], FAT [23], and CFA [35] are the state-of-the-art methods for improving the robust fairness of AT. FAT enhances robust fairness by controlling the variance of class-wise adversarial risk, which offers a distinct perspective compared to methods like FRL, CFA, and Fair-ARD. Hence, we mainly focus on comparing our Fair-ARD with both FRL and CFA, as they all employ the re-weighting approach. Firstly, they use different metrics to quantify the class difficulty. FRL and CFA use the class-wise robustness on the validation set, while we utilize the average distance between examples and the decision boundary, which offers a finer granularity, as shown in Fig. 2. Additionally, their re-weighting functions differ from ours. FRL's function is an iterative algorithm based on the difficulty metric, with more than four hyperparameters, and CFA applies a fractional function based on class difficulty, whereas we mainly use power functions for re-weighting. We make more discussions about several types of re-weighting functions in Appendix C.8. Finally, CFA utilizes the exponential moving average (EMA), while FRL and Fair-ARD do not employ it.

To show the advantages of our proposed method compared to FRL, FAT, and CFA, we also employed them in ARD, namely FRL-ARD, FAT-ARD, and CFA-ARD, respectively. The results in Table 4 demonstrate that while FRL and FAT can improve the worst-class robustness of ARD, they also lead to a slight decrease in the average robustness. On the other hand, Fair-ARD outperforms both FRL-ARD and FAT-ARD in terms of both average and worst-class robustness. Most notably, Fair-ARD improves the worst-class robustness under AA by 5.3% compared to FRL-ARD. This also provides rough evidence that the metric LPS is superior to FED. Regarding CFA, while CFA-ARD outperforms ARD in terms of robust fairness, it falls short when compared to Fair-ARD. This distinction highlights the superiority of Fair-ARD in enhancing robust fairness.

# 6 Conclusion

In this paper, we conducted an in-depth investigation into the issue of robust fairness in ARD. We revealed that the student model obtained through ARD fails to fully inherit the robust fairness from the teacher model. We discovered that this issue may arise due to the capacity gap between the teacher and student models, and previous ARD methods treated the knowledge of all classes equally. Based on these observations, we proposed Fair-ARD, which utilizes a refined class difficulty metric from a geometric perspective and a re-weighting strategy for distinct classes to enhance the robust fairness of the student model. Extensive experiments showed that Fair-ARD significantly improves the robust fairness of student models compared to existing methods while maintaining high overall robustness. Our work can help in establishing adversarially robust and fair lightweight deep learning models.

## Acknowledgements

This work was partially supported by the NSFC under Grants U20B2049, U21B2018, and 62302344, and the Fundamental Research Funds for the Central Universities, 2042023kf0122.

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
