# A   Algorithm

## A.1   Fair-ARD

Algorithm 1 is Fair Adversarial Robustness Distillation (Fair-ARD). Fair-ARD utilizes Eq.(5) to obtain adversarial data and geometric values. For each class, Fair-ARD reweights the distillation loss according to the average distance of its examples to the decision boundary and then updates the model parameters by minimizing the sum of the reweighted losses.

---

**Algorithm 1** Fair Adversarial Robustness Distillation (Fair-ARD)

---

**Input:** Student model $S$, teacher model $T$, training dataset $D = \{(x_i, y_i)\}_{i=1}^{n}$, number of classes $C$, learning rate $\eta$, number of epochs $N$, batch size $m$, number of batches $M$, temperature parameter $\tau$, the hyperparameter of ARD $\alpha$, smoothness hyperparameter $\beta$

**Output:** Adversarially robust and fair student model $S$

1: $\omega = \{1, \ldots, 1\}_1^{C}$
2: **for** epoch $= 1, \ldots, N$ **do**
3:      **for** mini-batch $= 1, \ldots, M$ **do**
4:          sample a mini-batch $\{(x_i, y_i)\}_{i=1}^{m}$ from $D$
5:          **for** $i = 1, \ldots, m$ (in parallel) **do**
6:              Obtain adversarial data $\tilde{x}_i$ of $x_i$ and geometry value $d(x_i, y_i)$ based on Eq. (5)
7:          **end for**
8:          $\theta \leftarrow \theta - \eta \nabla_\theta \left\{ \omega \left\{ (1-\alpha) \operatorname{CE}(S^\tau(x_i), y_i) + \alpha \tau^2 \operatorname{KL}(S^\tau(\tilde{x}_i), T^\tau(y_i)) \right\} \right\}$
9:      **end for**
10:      Compute $\omega$ for each class based on Eq.(6) and Eq.(7) with $\beta$.
11: **end for**

---

## A.2   Fair-IAD

Let $\omega_i$ be the weight for the $i$-th class; we formulate a Fair Introspective Adversarial Distillation (Fair-IAD) method as follows:

$$\min_{\theta_S} \frac{1}{C} \sum_{i=1}^{C} \frac{1}{n_i} \sum_{j=1}^{n_i} \omega_i \mathcal{L}_{\mathrm{IAD}}\left(S, T, x_i^j, y_i, \tau, \alpha\right), \tag{8}$$

where

$$\begin{aligned} &\mathcal{L}_{\mathrm{IAD}}\left(S, T, x_i^j, y_i, \tau, \alpha\right) \\ &= T_y\left(\tilde{x}_i^j\right)^\alpha \operatorname{KL}\left(S^\tau\left(\tilde{x}_i^j\right), T^\tau\left(x_i^j\right)\right) + \left(1 - T_y\left(\tilde{x}_i^j\right)^\alpha\right) \operatorname{KL}\left(S^\tau\left(\tilde{x}_i^j\right), S^\tau\left(x_i^j\right)\right), \end{aligned} \tag{9}$$

The method of IAD to generate adversarial examples is the same as that of ARD, and Eq. (5) is used to generate adversarial examples. Algorithm 2 is Fair Introspective Adversarial Distillation (Fair-IAD). Fair-IAD utilizes Eq. (5) to obtain adversarial data and geometric values.

## A.3   Fair-RSLAD

Let $\omega_i$ be the weight for the $i$-th class; we formulate a Fair Robust Soft Label Adversarial Distillation (Fair-RSLAD) method as follows:

$$\min_{\theta_S} \frac{1}{C} \sum_{i=1}^{C} \frac{1}{n_i} \sum_{j=1}^{n_i} \omega_i \mathcal{L}_{\mathrm{RSLAD}}\left(S, T, x_i^j, \alpha\right), \tag{10}$$

where

$$\mathcal{L}_{\mathrm{RSLAD}}\left(S, T, x_i^j, \alpha\right) = (1-\alpha) \operatorname{KL}\left(S\left(x_i^j\right), T\left(x_i^j\right)\right) + \alpha \operatorname{KL}\left(S\left(\tilde{x}_i^j\right), T\left(x_i^j\right)\right). \tag{11}$$

The method of RSLAD to generate adversarial examples is different from that of ARD, and Eq. (12) is used to generate adversarial examples. Algorithm 3 is Fair Robust Soft Label Adversarial Distillation

---

**Algorithm 2** Fair Introspective Adversarial Distillation (Fair-IAD)

---

**Input:** Student model $S$, teacher model $T$, training dataset $D = \{(x_i, y_i)\}_{i=1}^{n}$, number of classes $C$, learning rate $\eta$, number of epochs $N$, batch size $m$, number of batches $M$, temperature parameter $\tau$, the hyperparameter of IAD $\alpha$, smoothness hyperparameter $\beta$

**Output:** Adversarially robust and fair student model $S$

1: $\omega = \{1, \ldots, 1\}_1^C$
2: **for** epoch $= 1, \ldots, N$ **do**
3:      **for** mini-batch $= 1, \ldots, M$ **do**
4:          sample a mini-batch $\{(x_i, y_i)\}_{i=1}^{m}$ from $D$
5:          **for** $i = 1, \ldots, m$ (in parallel) **do**
6:              Obtain adversarial data $\tilde{x}_i$ of $x_i$ and geometry value $d(x_i, y_i)$ based on Eq. (5)
7:          **end for**

8:          $\theta \leftarrow \theta - \eta \nabla_\theta \left\{ \omega \left\{ \begin{array}{l} T_y(\tilde{x}_i)^\alpha \, \mathrm{KL}\left(S^\tau(\tilde{x}_i), T^\tau(x_i)\right) \\[1em] + \left(1 - T_y(\tilde{x}_i)^\alpha\right) \mathrm{KL}\left(S^\tau(\tilde{x}_i), S^\tau(x_i)\right) \end{array} \right\} \right\}$

9:      **end for**
10:     Compute $\omega$ for each class based on Eq. (6) and Eq. (7) with $\beta$.
11: **end for**

---

(Fair-RSLAD). Fair-RSLAD utilizes Eq. (12) to obtain adversarial data and geometric values.

$$\begin{cases} \tilde{x}^{(t+1)} = \Pi_{\mathcal{B}_\epsilon[x]}\left(\tilde{x}^{(t)} + \gamma \, \mathrm{sign}\left(\nabla_{\tilde{x}^{(t)}} \mathrm{KL}\left(S\left(\tilde{x}^{(t)}\right), T\left(\tilde{x}^{(t)}\right)\right)\right)\right), \\ d(x, y) = \underset{t \in [0, K]}{\mathrm{argmin}}\left(S\left(\tilde{x}^{(t)}\right) \neq y\right), \end{cases} \tag{12}$$

---

**Algorithm 3** Fair Robust Soft Label Adversarial Distillation (Fair-RSLAD)

---

**Input:** Student model $S$, teacher model $T$, training dataset $D = \{(x_i, y_i)\}_{i=1}^{n}$, number of classes $C$, learning rate $\eta$, number of epochs $N$, batch size $m$, number of batches $M$, the hyperparameter of RSLAD $\alpha$, smoothness hyperparameter $\beta$

**Output:** Adversarially robust and fair student model $S$

1: $\omega = \{1, \ldots, 1\}_1^C$
2: **for** epoch $= 1, \ldots, N$ **do**
3:      **for** mini-batch $= 1, \ldots, M$ **do**
4:          sample a mini-batch $\{(x_i, y_i)\}_{i=1}^{m}$ from $D$
5:          **for** $i = 1, \ldots, m$ (in parallel) **do**
6:              Obtain adversarial data $\tilde{x}_i$ of $x_i$ and geometry value $d(x_i, y_i)$ based on Eq. (12)
7:          **end for**
8:          $\theta \leftarrow \theta - \eta \nabla_\theta \left\{ \omega \left\{ (1-\alpha) \, \mathrm{KL}\left(S(x_i), T(x_i)\right) + \alpha \, \mathrm{KL}\left(S(\tilde{x}_i), T(x_i)\right) \right\} \right\}$
9:      **end for**
10:     Compute $\omega$ for each class based on Eq. (6) and Eq. (7) with $\beta$.
11: **end for**

---

## A.4 Fair-MTARD

Let $\omega_i$ be the weight for the $i$-th class; we formulate a Fair Multi-Teacher Adversarial Robustness Distillation (Fair-MTARD) method as follows:

$$\min_{\theta_S} \frac{1}{C} \sum_{i=1}^{C} \frac{1}{n_i} \sum_{j=1}^{n_i} \omega_i \mathcal{L}_{\mathrm{MTARD}}\left(S, T_{nat}, T_{adv}, x_i^j, \alpha\right), \tag{13}$$

where

$$\mathcal{L}_{\text{MTARD}}\left(S, T_{nat}, T_{adv}, x_i^j, \alpha\right)$$
$$= (1 - \alpha)\,\text{KL}\left(S\left(x_i^j\right), T_{nat}\left(x_i^j\right)\right) + \alpha\,\text{KL}\left(S\left(\tilde{x}_i^j\right), T_{adv}\left(x_i^j\right)\right), \tag{14}$$

where $T_{nat}$ is the natural teacher model, $T_{adv}$ is the robust teacher model.

The method of MTARD to generate adversarial examples is the same as that of ARD, and Eq. (5) is used to generate adversarial examples. Algorithm 4 is Fair Multi-Teacher Adversarial Robustness Distillation (Fair-MTARD). Fair-MTARD utilizes Eq. (5) to obtain adversarial data and geometric values.

---

**Algorithm 4** Fair Multi-Teacher Adversarial Robustness Distillation (Fair-MTARD)

---

**Input:** Student model $S$, natural teacher model $T_{nat}$, robust teacher model $T_{adv}$ training dataset
  $D = \{(x_i, y_i)\}_{i=1}^n$, number of classes $C$, learning rate $\eta$, number of epochs $N$, batch size $m$,
  number of batches $M$, the hyperparameter of MTARD $\alpha$, smoothness hyperparameter $\beta$
**Output:** Adversarially robust and fair student model $S$
 1: $\omega = \{1, \ldots, 1\}_1^C$
 2: **for** epoch $= 1, \ldots, N$ **do**
 3:     **for** mini-batch $= 1, \ldots, M$ **do**
 4:         sample a mini-batch $\{(x_i, y_i)\}_{i=1}^m$ from $D$
 5:         **for** $i = 1, \ldots, m$ (in parallel) **do**
 6:             Obtain adversarial data $\tilde{x}_i$ of $x_i$ and geometry value $d\left(x_i, y_i\right)$ based on Eq. (5)
 7:         **end for**
 8:         Update $\alpha$
 9:         $\theta \leftarrow \theta - \eta \nabla_\theta \left\{\omega \left\{(1 - \alpha)\,\text{KL}\left(S\left(x_i\right), T_{nat}\left(x_i\right)\right) + \alpha\text{KL}\left(S\left(\tilde{x}_i\right), T_{adv}\left(x_i\right)\right)\right\}\right\}$
10:     **end for**
11:     Compute $\omega$ for each class based on Eq. (6) and Eq. (7) with $\beta$.
12: **end for**

---

# B   Related Work

## B.1   Adversarial Attack

It has been observed that adding small perturbations to natural data to generate adversarial examples can lead to misclassifications in deep neural networks [31, 10]. Subsequently, a multitude of adversarial attack methods have been proposed. Adversarial attacks are divided into two types according to the information that the attacker can obtain. One is white-box attack, and the attacker can obtain all the information of the model, such as fast gradient sign method (FGSM) [10], projected gradient descent (PGD) [24], Carilini and Wagner attack (C&W) [4]; the other is a black-box attack, where the attacker can only obtain part of the output information of the model, including transfer-based attacks and query-based attacks. To provide a fuller evaluation of adversarial defenses, AutoAttack (AA) [8] was proposed, which is the most powerful attack so far.

## B.2   Adversarial Training

In the face of adversarial attacks, many defense methods have been proposed. Among them, adversarial training (AT) [24, 40, 34, 37, 41] has been proven to be the strongest defense method. AT enhances the model's robust generalization performance by incorporating adversarial examples into the training set. The objective function of AT can be summarized as follows:

$$\arg\min_{\theta} \mathcal{L}_{\min}\left(f_\theta\left(\tilde{x}\right), y\right), \text{ where } \tilde{x} = \arg\max_{\|\tilde{x} - x\|_p \leq \epsilon} \mathcal{L}_{\max}\left(f_\theta\left(\tilde{x}\right), y\right), \tag{15}$$

where $f_\theta$ is the DNN model with the parameter $\theta$, $\tilde{x}$ is the adversarial example of natural example $x$ within bounded $L_p$ distance $\epsilon$, $\mathcal{L}_{\min}$ is the outer minimization loss and $\mathcal{L}_{\max}$ is the the inner maximization loss. The inner maximization generates adversarial examples, and the outer minimization optimizes the model.

Geometry-Aware Instance-Reweighted Adversarial Training (GAIRAT) [41] introduces an adversarial training method that assigns varying weights to adversarial examples. The weights are determined by approximating the distance of each natural example to the decision boundary. The underlying premise of GAIRAT is that natural examples closer to the decision boundary are less robust, as their corresponding adversarial counterparts can more readily cross the decision boundary.

## B.3    Adversarial Robustness Distillation

Although AT is helpful for defending against adversarial examples, it is eager for large-capacity models (e.g., WideResNet-34-10 [39]), i.e., the larger the model capacity, the higher the robustness [1, 36, 45]. But for resource-constrained devices, small models (e.g. ResNet-18 [13], MobileNetV2 [30]) are more popular [30, 42, 32]. Therefore, adversarial robustness distillation (ARD) methods [9, 44, 45, 43] have been developed, leveraging both knowledge distillation and adversarial training techniques to augment the robustness of small models. Initially, [9] proposed ARD to transfer the robustness from a large model (teacher model) to improve the robustness of a small model (student model) by knowledge distillation. Following ARD, [44] proposed IAD, where student models trust their teacher models partially rather than fully; [45] proposed RSLAD, which leverages the robust soft labels of the teacher model to guide the learning of natural and adversarial examples in all loss terms; and [43] proposed MTARD, using a robust teacher and a natural teacher to jointly guide the learning of the student model. While these methods treat each class equally, we advocate class re-weighting to promote robust fairness.

## B.4    Robust Fairness

[38, 33] identified a significant disparity in the robustness of the adversarially trained model, which demonstrated high robustness on some classes while being exceedingly vulnerable on other classes, and [38] refers to this issue as *robust fairness*. [23] observed that with the increasing of the perturbation radius, stronger AT methods will lead to more severe robust fairness issues. Given the "buckets effect," the security of a system is typically predicated upon the robustness of its most vulnerable component. As such, even with an overall high robustness, a model's poor performance on a specific class of data may still introduce potential security threats. Considering that ARD is based on AT, it may also be susceptible to this inherent vulnerability.

Besides, [26] also introduced the concept of robustness bias, pointing out that different classes of data should have the same robustness, but did not propose specific implementation methods and did not consider the impact of knowledge distillation on robust fairness. Furthermore, the fairness metrics used in [26] differ from ours: [26] uses self-defined AUC, while we utilize the worst-class robustness and NSD.

In addition, we clarify the distinction between our notion of robust fairness and conventional machine learning (ML) fairness [12]. Firstly, robust fairness pertains to consistent predictive accuracy across classes under both standard and adversarial conditions [23]. In contrast, conventional ML fairness involves eliminating bias based on sensitive attributes like gender, race, etc. Secondly, prior works have studied fairness in classification, e.g., equalized odds [12]. However, to the best of our knowledge, we are the first to investigate the problem of achieving adversarially robust fairness through knowledge distillation.

## B.5    Fairness-oriented Distillation

In this subsection, we will clarify the difference between fairness in existing fairness-oriented distillation methods [18] and the fairness we are addressing.

First, their meanings differ. Prior work on fair distillation [18] aims to prevent discriminatory biases stemming from sensitive attributes. Our robust fairness focuses instead on minimizing differences in robustness across classes.

Second, their metrics for measuring fairness differ. [18] measures fairness via equalized odds metrics $DEO_M$ and $DEO_A$ which account for sensitive attributes. In contrast, our metrics of the worst-class accuracy and NSD are solely based on model performance across classes.

# C Extensive Experiments

## C.1 More Experimental Setup

In this subsection, we provide further details on the experiments conducted in Section 4.

**Adversarial Examples.** We use 10-step PGD (PGD-10) with a step size of $2/255$ to generate adversarial examples, and the training perturbation is bounded to the $L_\infty$ norm $\epsilon = 8/255$ for both datasets.

**Teachers' Performance.** Table 5 reports the average robustness and worst-class robustness of the teacher models under different attacks.

**Experimental Environment.** All experiments are run on NVIDIA RTX 3090, utilizing the PyTorch framework.

Table 5: The average (Avg.) and worst-class (Worst) robustness of the teacher models used in our experiments. RN: ResNet; PRN:PreActResNet; WRN: WideResNet.

| Dataset | Model | Type | Clean | | FGSM | | PGD | | C&W | | AA | |
|---|---|---|---|---|---|---|---|---|---|---|---|---|
| | | | Avg. | Worst | Avg. | Worst | Avg. | Worst | Avg. | Worst | Avg. | Worst |
| CIFAR-10 | RN-56 | Natural | 92.72 | 84.40 | 8.23 | 1.60 | 0.00 | 0.00 | 0.00 | 0.00 | 0.00 | 0.00 |
| | WRN-34-10 | Robust | 84.92 | 67.00 | 61.11 | 33.00 | 55.32 | 26.80 | 53.91 | 24.40 | 52.55 | 23.30 |
| CIFAR-100 | WRN-22-6 | Natural | 76.67 | 48.00 | 5.03 | 0.00 | 0.00 | 0.00 | 0.00 | 0.00 | 0.00 | 0.00 |
| | WRN-70-16 | Robust | 60.96 | 24.00 | 35.92 | 6.00 | 33.56 | 5.00 | 31.12 | 3.00 | 28.80 | 6.00 |
| SVHN | WRN-34-10 | Robust | 94.53 | 88.92 | 91.80 | 80.60 | 59.48 | 34.22 | 58.60 | 33.61 | 57.24 | 31.75 |
| Tiny-ImageNet | PRN-18 | Robust | 40.58 | 16.85 | 19.80 | 3.10 | 18.18 | 2.50 | 13.90 | 1.10 | 13.45 | 1.00 |

## C.2 More Results of Section 2

In this subsection, we present the complete results of Table 1 in Table 6. As observed from the results in Table 6, easier classes (those with higher robustness in the teacher model) exhibit a higher proportion of robustness inheritance, implying that the student model is more inclined towards these easier classes.

## C.3 Experiments on CIFAR100

In this subsection, we provide the experimental results for CIFAR100. The experimental setup for CIFAR100 is essentially the same as that for CIFAR10, except for differences in the teacher model. Moreover, we cannot reproduce the ARD [9] results on CIFAR100; hence, no ARD-related experiments are conducted on CIFAR100.

**Teacher-Student Pairs.** Following [45, 43], we consider two student networks, ResNet-18 [13] and MobileNetV2 [30], and a robust teacher network WideResNet-70-16 provided by [11], and a natural teacher network WideResNet-22-6 [39]. Natural teacher networks trained with cross-entropy loss are used by MTARD. The average and worst-class robustness of teacher models are in Table 5.

**Results.** We present the average robustness and worst-class robustness of various algorithms in Table 7 and Table 8, and the NSD of vanilla ARD and Fair-ARD in Fig. 6 and Fig. 7. The results show that, in alignment with the experimental results on CIFAR10, our Fair-ARD manifests a distinct superiority in augmenting the robust fairness of the student model on CIFAR100.

## C.4 Experiments on Other Datasets

In this subsection, we provide the experimental results for SVHN and Tiny-ImageNet. The experimental setup for SVHN and Tiny-ImageNet is essentially the same as that for CIFAR10, except for differences in the teacher model. Moreover, since the worst-class robustness is extremely low and there are only 50 images for each class in the test set, we report the average of the worst-20% class robustness on Tiny-ImageNet following [35].

Table 6: The robustness of the teacher model WideResNet-34-10 on each class and the percentage of the robustness for each class inherited by the student model ResNet-18 with respect to the corresponding class robustness of the teacher model. The dataset is CIFAR10.

| Attack | Method | plane | car | bird | cat | deer | dog | frog | horse | ship | truck |
|--------|--------|-------|-----|------|-----|------|-----|------|-------|------|-------|
| | Teacher | 68.50 | 83.10 | 44.60 | 33.00 | 54.70 | 44.70 | 64.50 | 68.30 | 75.10 | 74.60 |
| FGSM | ARD | 94.31% | 94.95% | 88.34% | 85.15% | 88.30% | 99.78% | 94.11% | 100.73% | 105.46% | 100.67% |
| | IAD | 100.29% | 96.39% | 97.76% | 84.55% | 84.83% | 101.79% | 96.74% | 102.78% | 96.40% | 96.25% |
| | RSLAD | 102.48% | 95.07% | 95.74% | 90.30% | 95.80% | 102.01% | 96.90% | 99.56% | 101.73% | 98.79% |
| | MTARD | 102.48% | 99.28% | 101.79% | 89.70% | 88.30% | 105.59% | 102.02% | 105.12% | 95.34% | 100.94% |
| | Teacher | 63.40 | 79.00 | 40.10 | 27.00 | 43.40 | 41.30 | 57.20 | 63.60 | 69.20 | 69.20 |
| PGD | ARD | 93.85% | 92.78% | 80.80% | 80.60% | 82.49% | 94.48% | 90.02% | 100.32% | 103.02% | 98.70% |
| | IAD | 99.05% | 94.42% | 94.76% | 81.72% | 82.03% | 98.08% | 93.87% | 102.21% | 92.23% | 93.77% |
| | RSLAD | 103.00% | 93.79% | 94.01% | 88.81% | 98.39% | 99.52% | 95.62% | 100.32% | 102.01% | 97.54% |
| | MTARD | 98.42% | 96.70% | 89.28% | 75.37% | 70.28% | 93.53% | 88.44% | 101.74% | 87.34% | 94.64% |
| | Teacher | 62.90 | 78.00 | 38.40 | 24.40 | 40.60 | 40.70 | 53.70 | 62.40 | 68.50 | 69.50 |
| C&W | ARD | 95.23% | 93.59% | 80.99% | 80.74% | 83.50% | 97.30% | 95.34% | 101.44% | 105.26% | 98.85% |
| | IAD | 100.16% | 95.26% | 94.53% | 86.89% | 76.85% | 98.77% | 96.09% | 103.69% | 93.14% | 91.80% |
| | RSLAD | 102.54% | 94.36% | 93.49% | 92.62% | 95.32% | 100.25% | 95.34% | 100.32% | 102.34% | 97.27% |
| | MTARD | 97.93% | 96.41% | 88.02% | 71.31% | 67.49% | 92.38% | 87.90% | 101.28% | 87.88% | 91.94% |
| | Teacher | 62.40 | 77.20 | 37.30 | 23.30 | 37.60 | 39.40 | 51.60 | 61.80 | 67.10 | 67.80 |
| AA | ARD | 93.75% | 93.13% | 78.55% | 72.53% | 81.38% | 95.43% | 90.50% | 100.00% | 104.92% | 98.53% |
| | IAD | 98.24% | 94.69% | 92.49% | 79.83% | 74.47% | 96.95% | 91.86% | 102.10% | 92.55% | 93.22% |
| | RSLAD | 101.28% | 94.56% | 93.57% | 89.70% | 96.01% | 99.75% | 94.38% | 99.19% | 102.38% | 96.90% |
| | MTARD | 95.35% | 96.11% | 84.99% | 63.52% | 60.37% | 88.58% | 78.29% | 98.54% | 83.61% | 89.38% |

Table 7: The average (Avg.) and worst-class (Worst) robustness for various algorithms using ResNet18 on CIFAR-100. Better results in comparison with vanilla ARD and our proposed fair version are **bolded**.

| Method | Clean | | FGSM | | PGD | | C&W | | AA | |
|--------|-------|-------|-------|-------|-------|-------|-------|-------|-------|-------|
| | Avg. | Worst | Avg. | Worst | Avg. | Worst | Avg. | Worst | Avg. | Worst |
| Natural | 75.32 | 46.00 | 8.29 | 0.00 | 0.01 | 0.00 | 0.00 | 0.00 | 0.00 | 0.00 |
| SAT | 57.81 | 18.00 | 29.09 | 5.00 | 24.07 | 2.00 | 23.69 | 2.00 | 21.80 | 2.00 |
| TRADES | 54.10 | 16.00 | 30.20 | 4.00 | 27.70 | 3.00 | 24.23 | 3.00 | 23.37 | 2.00 |
| IAD | 55.76 | 16.00 | 32.43 | 2.00 | **29.38** | 2.00 | 26.73 | 2.00 | 25.04 | 1.00 |
| Fair-IAD (ours) | **57.08** | **18.00** | **32.87** | **7.00** | **29.38** | **6.00** | **26.92** | **3.00** | **25.55** | **2.00** |
| RSLAD | 58.06 | 15.00 | 34.33 | **5.00** | 30.82 | 3.00 | 28.30 | 2.00 | **26.57** | 2.00 |
| Fair-RSLAD (ours) | **58.33** | **21.00** | **34.48** | **5.00** | **30.94** | **5.00** | **28.37** | **4.00** | 26.37 | **3.00** |
| MTARD | **64.34** | 4.00 | 30.07 | 0.00 | 23.51 | 0.00 | 22.95 | 0.00 | 20.16 | 0.00 |
| Fair-MTARD (ours) | 57.72 | **22.00** | **33.72** | **6.00** | **30.39** | **5.00** | **27.85** | **3.00** | **26.18** | **3.00** |

**Teacher-Student Pairs.** Following [45, 43], we adopt ResNet-18 as the student network. For SVHN, a robust WideResNet34-10, trained via TRADES, serves as the teacher network. For Tiny-ImageNet, we use a robust teacher network PreActResNet-18 [14] provided by [17]. The average and worst-class robustness of teacher models are in Table 5.

**Results.** As the Table 9 shows, compared to IAD, our Fair-ARD framework improved worst-class (worst-20% class) robustness under AA by 3.98% (0.45%) on SVHN (Tiny-ImageNet). These consistent enhancements on additional benchmark datasets further demonstrate the generality of our method.

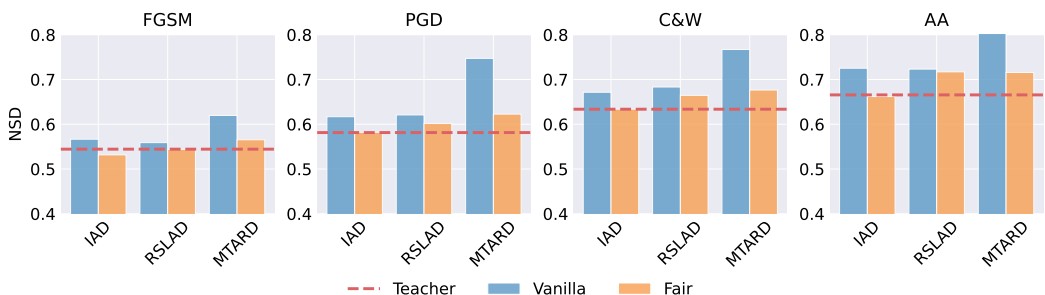

Figure 6: The NSD of the vanilla ARD (Vanilla) and our proposed fair version (Fair) using ResNet18 on CIFAR-100. The red line represents the NSD of the teacher model (WideResNet34-10). From left to right, the panels show results under FGSM, PGD, C&W, and AA, respectively.

Table 8: The average (Avg.) and worst-class (Worst) robustness for various algorithms using MobileNetV2 on CIFAR-100. Better results in comparison with vanilla ARD and our proposed fair version are **bolded**.

| Method | Clean | | FGSM | | PGD | | C&W | | AA | |
|---|---|---|---|---|---|---|---|---|---|---|
| | Avg. | Worst | Avg. | Worst | Avg. | Worst | Avg. | Worst | Avg. | Worst |
| Natural | 74.56 | 45.00 | 6.83 | 0.00 | 0.00 | 0.00 | 0.00 | 0.00 | 0.00 | 0.00 |
| SAT | 56.67 | 7.00 | 31.61 | 0.00 | 28.28 | 0.00 | 26.60 | 0.00 | 24.43 | 0.00 |
| TRADES | 57.03 | 15.00 | 31.87 | 3.00 | 29.15 | 2.00 | 24.78 | 1.00 | 23.96 | 1.00 |
| IAD | **55.96** | 14.00 | 32.09 | 1.00 | 29.18 | 1.00 | **27.02** | 1.00 | 25.14 | 1.00 |
| Fair-IAD (ours) | 55.45 | **16.00** | **32.11** | **6.00** | **29.29** | **6.00** | 26.89 | **2.00** | **25.25** | **2.00** |
| RSLAD | 58.43 | 15.00 | 34.00 | 3.00 | 30.05 | 2.00 | 27.88 | 1.00 | 25.97 | 0.00 |
| Fair-RSLAD (ours) | **59.01** | **20.00** | **34.12** | **4.00** | **30.47** | **4.00** | **28.11** | **3.00** | **26.36** | **1.00** |
| MTARD | 52.62 | 0.00 | 27.90 | 0.00 | 22.69 | 0.00 | 22.11 | 0.00 | 19.35 | 0.00 |
| Fair-MTARD (ours) | **58.46** | **22.00** | **33.86** | **6.00** | **29.88** | **5.00** | **27.88** | **3.00** | **25.93** | **2.00** |

## C.5  Standard Deviation

All training runs were independently conducted five times and reported averages, omitting negligible standard deviations. We have presented the results of using ResNet18 on CIFAR100 in Table 10. These minor standard deviations align with prior AT work [36, 22], further demonstrating the stability.

## C.6  Stability of Fair-ARD

Fig. 8 presented the variations in the weights of each class during training. And we found that the weights initially fluctuated but gradually stabilized in the later stages.

## C.7  Different PGD Step Sizes

We analyze the sensitivity of the step size theoretically and empirically as follows.

Increasing step size $\gamma$ reduces steps for examples to cross decision boundaries, diminishing LPS discrimination. As shown in Table 11, $\gamma = 3/255$ versus $2/255$ maintains average robustness but worsens worst-class due to weakened re-weighting from reduced LPS discrimination. However, Fair-ARD still exceeds ARD.

Conversely, decreasing $\gamma$ can raise steps to boundaries, potentially improving discrimination. But smaller $\gamma$ risks lower quality adversarial examples for training, hindering distillation. As shown in Table 11, a 10x smaller $\gamma$ substantially decreased robustness, which can be attributed to poorer adversarial examples.

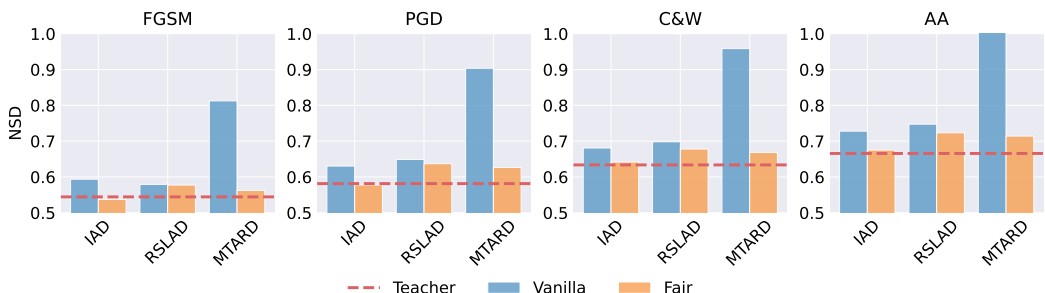

Figure 7: The NSD of the vanilla ARD (Vanilla) and our proposed fair version (Fair) using MobileNetV2 on CIFAR-100. The red line represents the NSD of the teacher model (WideResNet34-10). From left to right, the panels show results under FGSM, PGD, C&W, and AA, respectively.

Table 9: The average (Avg.) and worst-class (Worst) robustness for various algorithms using ResNet18 on SVHN and Tiny-ImageNet. Better results in comparison with vanilla ARD and our proposed fair version are **bolded**.

| Dataset | Method | Clean | | FGSM | | PGD | | CW | | AA | |
|---------|--------|-------|-------|-------|-------|-------|-------|-------|-------|-------|-------|
| | | Avg. | Worst | Avg. | Worst | Avg. | Worst | Avg. | Worst | Avg. | Worst |
| SVHN | IAD | 90.28 | 74.04 | **81.61** | 53.98 | 58.19 | 25.84 | 57.31 | 25.48 | 43.58 | 20.12 |
| | Fair-IAD (ours) | **91.66** | **83.19** | 80.39 | **61.87** | **68.08** | **41.14** | **66.96** | **40.78** | **44.55** | **24.10** |
| Tiny-ImageNet | IAD | 40.07 | 15.95 | 21.04 | 3.80 | 19.33 | 2.85 | 15.35 | 1.35 | 14.46 | 1.15 |
| | Fair-IAD (ours) | **40.94** | **16.80** | **21.90** | **4.65** | **20.09** | **3.90** | **16.08** | **1.85** | **15.35** | **1.60** |

In summary, $\gamma$ tuning balances discrimination versus adversarial training efficacy. Careful adjustment optimizes Fair-ARD performance.

## C.8 Different Re-weighting Functions

In this subsection, to gain a more comprehensive understanding of Fair-ARD, we illustrate the influence of different re-weighting functions on Fair-ARD. Specifically, we choose ResNet-18 as the student model, ARD as the baseline method, and conduct the experiment on CIFAR10. Other settings remain consistent with Section 4.1.

In the left panel of Fig. 9, besides power-type Eq. (7) (purple line), which we have consistently employed across all other experiments, we compare different types of re-weighting functions following GAIRAT [41]. The orange line represents a linear-type function, i.e.,

$$\omega_i = 1 - \frac{\kappa_i}{K + 1}, \tag{16}$$

where $K$ is the PGD steps.

The green line represents a sigmoid-type function, i.e.,

$$\omega_i = \sigma(\lambda + 5 \times (1 - 2 \times \kappa_i/K)), \tag{17}$$

where $\sigma(x) = \frac{1}{1+e^{-x}}$, $\lambda$ is the parameter. Following [41], we adopt $\lambda = 0$.

The red line represents a tanh-type function, i.e.,

$$\omega_i = \frac{(1 + \tanh(\lambda + 5 \times (1 - 2 \times \kappa_i/K)))}{2}, \tag{18}$$

where $\lambda$ is the parameter. Following [41], we adopt $\lambda = 0$.

We present the average robustness and worst-class robustness of Fair-ARD using different re-weighting functions in Table 12, and their NSD in the right panel of Fig. 9. The results show that Fair-ARD with the power-type function (Eq. (7)) achieves the best average and worst-case class robustness, while Fair-ARD with the sigmoid-type function (Eq. (17)) has the optimal NSD.

Table 10: The average (Avg.) and worst-class (Worst) robustness and their standard deviations for various algorithms using ResNet18 on CIFAR-100.

| Method | Clean | | PGD | | AA | |
|---|---|---|---|---|---|---|
| | Avg. | Worst | Avg. | Worst | Avg. | Worst |
| Natural | 75.32±0.65 | 46.00±0.47 | 0.01±0.00 | 0.00±0.00 | 0.00±0.00 | 0.00±0.00 |
| SAT | 57.81±0.61 | 18.00±0.55 | 24.07±0.56 | 2.00±0.41 | 21.80±0.47 | 2.00±0.41 |
| TRADES | 54.10±0.68 | 16.00±0.72 | 27.70±0.51 | 3.00±0.43 | 23.37±0.54 | 2.00±0.17 |
| IAD | 55.76±0.64 | 16.00±0.47 | **29.38±0.41** | 2.00±0.64 | 25.04±0.53 | 1.00±0.71 |
| Fair-IAD (ours) | **57.08±0.34** | **18.00±0.33** | **29.38±0.56** | **6.00±0.62** | **25.55±0.26** | **2.00±0.65** |
| RSLAD | 58.06±0.51 | 15.00±0.45 | 30.82±0.66 | 3.00±0.52 | **26.57±0.62** | 2.00±0.55 |
| Fair-RSLAD (ours) | **58.33±0.45** | **21.00±0.48** | **30.94±0.53** | **5.00±0.51** | 26.37±0.51 | **3.00±0.51** |
| MTARD | **64.34±0.28** | 4.00±0.43 | 23.51±0.58 | 0.00±0.00 | 20.16±0.73 | 0.00±0.00 |
| Fair-MTARD (ours) | 57.72±0.21 | **22.00±0.41** | **30.39±0.46** | **5.00±0.43** | **26.18±0.53** | **3.00±0.43** |

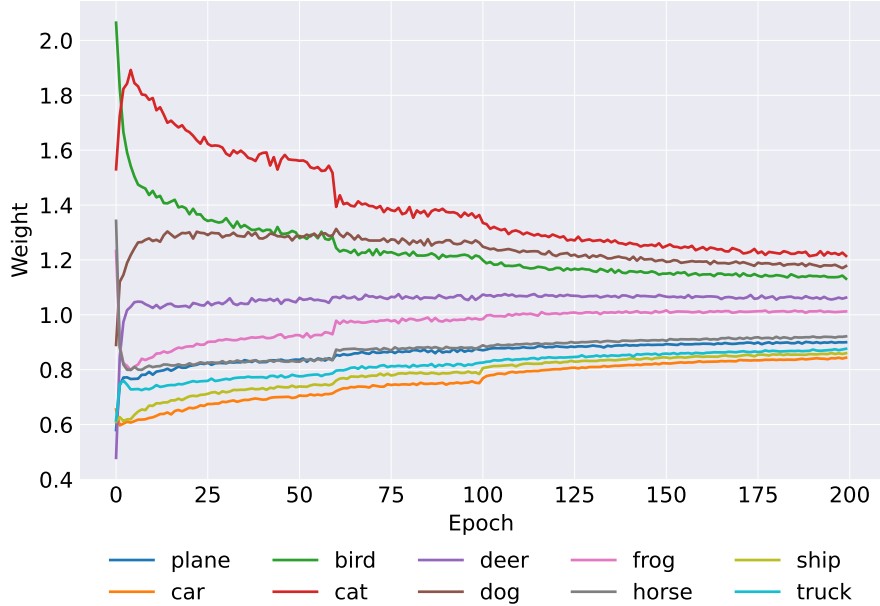

Figure 8: The weights of each class in Fair-ARD using ResNet18 on CIFAR-10 across epochs.

## C.9 Different Teachers

In this subsection, to further verify the effectiveness of Fair-ARD, we conduct experiments using different teacher models. Specifically, we choose ResNet-18 as the student model, with ARD as the baseline method, and perform the experiments on CIFAR10. Other settings remain consistent with Section 4.1. Table 13 presents the average robustness and worst-class robustness for the different teacher models.

We present the average robustness and worst-class robustness of Fair-ARD using different teacher models in Table 14 and their NSD in Fig. 10. The results show that, regardless of the teacher model used, our Fair-ARD demonstrates superiority in improving the robust fairness of the student model.

Interestingly, we observe that the average robustness and the worst-class robustness of the student model do not monotonically increase with the robustness of the teacher model. The performance of the student model under the guidance of WRN70-16 is inferior to that of WRN34-20 and even WRN34-10. This aligns with the observations in [45], where the robustness of the student model tends to decrease when the teacher model becomes overwhelmingly complex and unlearnable for the student model. These findings indicate that selecting an appropriately large teacher model also poses an unresolved challenge, a direction we aim to explore in our future work.

Table 11: The average (Avg.) and worst-class (Worst) robustness for various algorithms and step sizes using ResNet18 on CIFAR-10. The best results are **bolded**, and the second best results are underlined.

| Method | Step Size | Clean | | FGSM | | PGD | | C&W | | AA | |
|---|---|---|---|---|---|---|---|---|---|---|---|
| | | Avg. | Worst | Avg. | Worst | Avg. | Worst | Avg. | Worst | Avg. | Worst |
| ARD | 2/255 | 83.22 | 61.00 | 58.77 | 28.10 | 51.65 | 21.60 | **51.25** | 19.70 | 49.05 | 16.90 |
| Fair-ARD (ours) | 0.2/255 | **86.21** | **70.30** | 54.28 | 30.40 | 39.69 | 18.90 | 39.79 | 18.80 | 36.39 | 14.90 |
| Fair-ARD (ours) | 1/255 | 84.36 | 69.90 | 58.51 | 36.10 | 50.86 | 27.90 | 50.33 | 26.80 | 47.95 | 22.80 |
| Fair-ARD (ours) | 2/255 | 82.96 | 68.10 | 57.69 | 39.20 | **52.05** | **33.20** | 50.69 | **31.00** | **49.13** | **29.20** |
| Fair-ARD (ours) | 3/255 | 83.90 | 69.60 | **58.80** | **41.00** | 51.89 | 31.80 | 50.76 | 29.60 | 49.10 | 27.00 |

Table 12: The average (Avg.) and worst-class (Worst) robustness for Fair-ARD with different re-weighting functions using ResNet18 on CIFAR-10. The best results are **bolded**, and the second best results are underlined.

| Function | Clean | | FGSM | | PGD | | C&W | | AA | |
|---|---|---|---|---|---|---|---|---|---|---|
| | Avg. | Worst | Avg. | Worst | Avg. | Worst | Avg. | Worst | Avg. | Worst |
| w/o | **83.22** | 61.00 | **58.77** | 28.10 | 51.65 | 21.60 | **51.25** | 19.70 | 49.05 | 16.90 |
| linear | 82.69 | **68.60** | 58.23 | 38.70 | 51.75 | 31.60 | 50.73 | 29.10 | 48.97 | 27.00 |
| sigmoid | 82.91 | 67.90 | 57.87 | 39.10 | 51.59 | 31.10 | 50.16 | 28.30 | 48.58 | 26.60 |
| tanh | 76.90 | 55.70 | 51.08 | 28.50 | 46.22 | 24.80 | 45.04 | 22.90 | 43.62 | 22.60 |
| power | 82.96 | 68.10 | 57.69 | **39.20** | **52.05** | **33.20** | 50.69 | **31.00** | **49.13** | **29.20** |

## C.10 More Discussions

In this subsection, to validate the effectiveness of our proposed fairness framework in AT, we apply our proposed re-weighting strategy to TRADES [40] and then compare it with GAIRAT [41], Fair Robust Learning (FRL) [38], and Fair Adversarial Training (FAT) [23].

Let $\omega_i$ be the weight for the $i$-th class, we formulate a Fair-TRADES method as follows:

$$\min_{\theta_f} \frac{1}{C} \sum_{i=1}^{C} \frac{1}{n_i} \sum_{j=1}^{n_i} \omega_i \mathcal{L}_{\text{TRADES}} \left( f, x_i^j, y, \beta' \right), \tag{19}$$

where

$$\mathcal{L}_{\text{TRADES}} \left( f, x_i^j, y, \beta' \right) = \text{CE} \left( f \left( x_i^j \right), y \right) + \beta' \, \text{KL} \left( f \left( \tilde{x}_i^j \right), f \left( x_i^j \right) \right), \tag{20}$$

where $\mathcal{L}_{\text{TRADES}}$ is the loss of TRADES, $f$ is the DNN model with the parameter $\theta_f$, $\beta'$ is the hyperparameter controlling the weight of the adversarial loss. The method of TRADES to generate adversarial examples is different from that of ARD, and Eq. (21) is used to generate adversarial examples.

$$\begin{cases} \tilde{x}^{(t+1)} = \Pi_{\mathcal{B}_\epsilon[x]} \left( \tilde{x}^{(t)} + \gamma \, \text{sign} \left( \nabla_{\tilde{x}^{(t)}} \, \text{KL} \left( f \left( \tilde{x}^{(t)} \right), f \left( x^{(t)} \right) \right) \right) \right) \\ d(x, y) = \underset{t \in [0, K]}{\text{argmin}} \left( f \left( \tilde{x}^{(t)} \right) \neq y \right), \end{cases} \tag{21}$$

For an equitable comparison, we apply the techniques of GAIRAT, FRL, and FAT to TRADES, thereby forming GAIR-TRADES, FRL-TRADES, and FAT-TRADES, respectively, while maintaining the same parameters as in their original papers. The results of these algorithms compared with Fair-TRADES are presented in Table 15. From the results, GAIR-TRADES performs exceptionally well under FGSM and PGD attacks, yet its robustness significantly diminishes when facing C&W and AA, performing even poorer than vanilla TRADES. Although FRL-TRADES and FAT-TRADES improve the worst-class robustness compared to vanilla TRADES, they noticeably decrease the average robustness. Meanwhile, Fair-TRADES improves the worst-class robustness under all attacks and maintains high average robustness, arguably making it the best method in terms of overall performance among these algorithms.

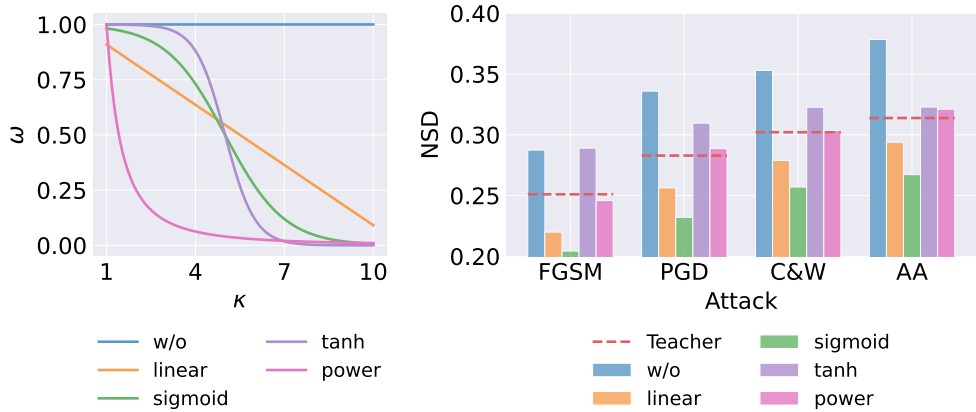

Figure 9: Effects of different re-weighting functions on Fair-ARD using ResNet18 on CIFAR10. *Left*: $\omega$ values under different re-weighting functions. *Right*: the NSD of Fair-ARD with different re-weighting functions. The red lines represent the NSD of the teacher model (WideResNet34-10) under different attacks.

Table 13: The average (Avg.) and worst-class (Worst) robustness of different teacher models on CIFAR-10. RN: ResNet; WRN: WideResNet.

| Model | Clean | | FGSM | | PGD | | C&W | | AA | |
|---|---|---|---|---|---|---|---|---|---|---|
| | Avg. | Worst | Avg. | Worst | Avg. | Worst | Avg. | Worst | Avg. | Worst |
| RN-18 | 82.22 | 64.80 | 58.38 | 31.00 | 52.35 | 25.80 | 50.33 | 22.30 | 49.01 | 21.20 |
| RN-34 | 84.12 | 70.10 | 60.20 | 34.50 | 52.66 | 26.20 | 51.74 | 24.50 | 49.73 | 22.20 |
| WRN-34-10 | 84.92 | 67.00 | 61.11 | 33.00 | 55.32 | 26.80 | 53.91 | 24.40 | 52.55 | 23.30 |
| WRN-34-20 | 85.65 | 71.60 | 64.48 | 38.80 | 59.84 | 33.60 | 57.80 | 29.30 | 56.85 | 28.40 |
| WRN-70-16 | 85.29 | 72.30 | 64.40 | 38.20 | 59.59 | 32.50 | 58.26 | 30.70 | 57.17 | 29.20 |

### C.11  Computational Cost Comparison

In this subsection, we compare the computational cost of vanilla ARD and Fair-ARD. Specifically, we use WideResNet34-10 as the teacher model and ResNet18 as the student model, conducting experiments on CIFAR10. We present the average training time per epoch and GPU memory consumption of different algorithms in Table 16. The results show that Fair-ARD requires more time than vanilla ARD due to the need to calculate class difficulty metrics and implement re-weighting strategies. However, Fair-ARD and vanilla ARD consume the same amount of GPU memory, as the additional memory overhead introduced by Fair-ARD is negligible.

## D  Broader Impacts

In safety-critical applications, deploying robust lightweight Deep Neural Networks (DNNs) with robustness biases toward different classes of data can pose significant risks. For instance, if DNNs in autonomous vehicles exhibit high robustness to inanimate objects on the road while being less robust to pedestrians, they could potentially place pedestrians in dangerous situations when facing adversarial attacks. In this paper, our Fair Adversarial Robustness Distillation (Fair-ARD) framework can mitigate this safety concern. By promoting robust and fair lightweight models, our framework enhances the comprehensive safety of DNNs deployed in resource-constrained and safety-critical devices.

Moreover, in many areas sensitive to social ethics, it is crucial to ensure that DNNs do not exhibit discriminatory behaviors towards certain classes (groups or populations). Our framework also offers valuable contributions in this realm of research.

Table 14: The average (Avg.) and worst-class (Worst) robustness for vanilla ARD and Fair-ARD using ResNet18 with different teacher models on CIFAR10. Better results in comparison with vanilla ARD and our proposed fair version are **bolded**. RN: ResNet; WRN: WideResNet.

| Teacher | Method | Clean | | FGSM | | PGD | | C&W | | AA | |
|---|---|---|---|---|---|---|---|---|---|---|---|
| | | Avg. | Worst | Avg. | Worst | Avg. | Worst | Avg. | Worst | Avg. | Worst |
| RN-18 | ARD | 81.36 | 59.30 | **56.81** | 27.80 | 50.45 | 20.80 | **49.49** | 20.10 | **47.83** | 17.00 |
| | Fair-ARD (ours) | **81.85** | **66.80** | **56.81** | **33.70** | **50.83** | **26.90** | 49.37 | **24.80** | 47.76 | **22.30** |
| RN-34 | ARD | **80.63** | 59.80 | **56.08** | 25.40 | 50.53 | 20.40 | **49.49** | 19.10 | 47.81 | 17.20 |
| | Fair-ARD (ours) | 80.31 | **65.00** | 56.06 | **34.70** | **50.78** | **29.00** | 49.31 | **27.70** | **47.82** | **24.10** |
| WRN34-10 | ARD | **83.22** | 61.00 | **58.77** | 28.10 | 51.65 | 21.60 | **51.25** | 19.70 | 49.05 | 16.90 |
| | Fair-ARD (ours) | 82.96 | **68.10** | 57.69 | **39.20** | **52.05** | **33.20** | 50.69 | **31.00** | **49.13** | **29.20** |
| WRN34-20 | ARD | **82.28** | 62.70 | **58.47** | 30.80 | 52.10 | 24.40 | **50.57** | 23.20 | **48.85** | 19.40 |
| | Fair-ARD (ours) | 81.87 | **66.20** | 57.74 | **41.30** | **52.11** | **34.30** | 50.26 | **32.00** | 48.67 | **28.60** |
| WRN70-16 | ARD | **82.52** | 61.30 | 57.76 | 28.50 | 51.73 | 23.40 | 50.09 | 20.10 | 48.45 | 18.10 |
| | Fair-ARD (ours) | 81.80 | **66.50** | **58.03** | **38.00** | **52.01** | **30.70** | **50.47** | **29.30** | **48.91** | **26.20** |

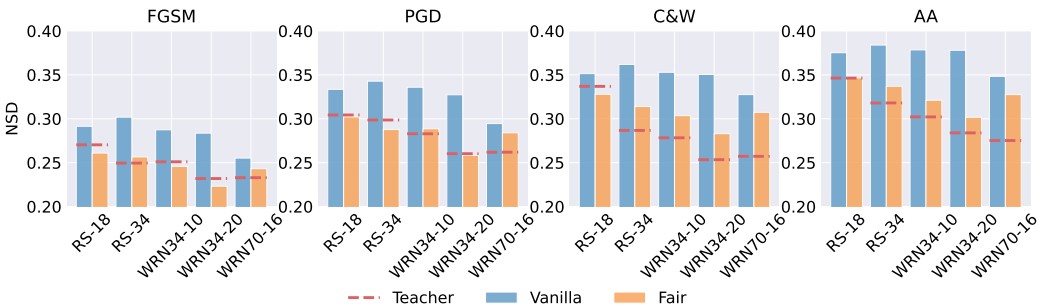

Figure 10: The NSD of the vanilla ARD (Vanilla) and our proposed fair version (Fair) using ResNet18 with different teacher models on CIFAR-10. The red lines represent the NSD of the teacher models under different attacks. From left to right, the panels show results under FGSM, PGD, C&W, and AA, respectively.

## E    Limitations

Firstly, the re-weighting function of Fair-ARD is heuristic and may not be optimal. Given the diverse perspectives in AT, there may exist more effective methods. Besides, our adopted re-weighting algorithm is an intuitive design. More fine-grained re-weighting methods may achieve higher worst-class robustness.

In addition, our Fair-ARD, like other existing ARD methods, is based on offline distillation, where the student model learns from a pretrained, fixed-parameter teacher model. Offline distillation, however, does not guarantee that the learning processes of the teacher and student models are matched, nor does it allow the teacher's knowledge teaching process to be adjusted in real-time based on the learning state of the student model. If there is a significant gap in performance between the fully trained teacher model and the student model, this may affect the student's learning in the initial stage. In contrast, online distillation is a mode in which both the teacher and student models participate in training and parameter updates simultaneously. In the process of online distillation, both the teacher and student models update their parameters synchronously. Therefore, the mode of distillation may be a factor constraining the performance of the student model. We leave a detailed study of the impact of distillation modes on robust fairness for future work.

Table 15: The average (Avg.) and worst-class (Worst) robustness for various algorithms using ResNet18 on CIFAR-10. The best results are **bolded**, and the second best results are underlined.

| Method | Clean | | FGSM | | PGD | | C&W | | AA | |
|---|---|---|---|---|---|---|---|---|---|---|
| | Avg. | Worst | Avg. | Worst | Avg. | Worst | Avg. | Worst | Avg. | Worst |
| TRADES | 82.22 | 64.80 | 58.38 | 31.00 | 52.35 | 25.80 | **50.33** | 22.30 | **49.01** | 21.20 |
| GAIR-TRADES | **82.37** | 66.00 | **59.35** | **42.90** | **54.78** | **38.80** | 43.19 | 16.20 | 39.66 | 12.10 |
| FRL-TRADES | 82.20 | **69.40** | 55.78 | 36.30 | 48.82 | 29.60 | 47.14 | 26.80 | 45.51 | 24.90 |
| FAT-TRADES | 79.92 | 66.30 | 56.19 | 35.20 | 51.76 | 30.20 | 48.87 | 23.80 | 47.67 | 22.10 |
| Fair-TRADES | 82.29 | 68.50 | 57.73 | 38.20 | 52.00 | 32.00 | 49.55 | **27.40** | 48.20 | **25.90** |

Table 16: Time and GPU Memory cost of vanilla ARD and Fair-ARD using ResNet18 on CIFAR-10.

| Method | Time (Avg. Epoch) | GPU Memory |
|---|---|---|
| ARD | 130.14s | 3645MiB |
| Fair-ARD (ours) | 154.96s | 3645MiB |
| IAD | 167.22s | 4231MiB |
| Fair-IAD (ours) | 207.70s | 4231MiB |
| RSLAD | 161.50s | 3877MiB |
| Fair-RSLAD (ours) | 215.23s | 3877MiB |
| MTARD | 229.96s | 4017MiB |
| Fair-MTARD (ours) | 232.87s | 4017MiB |