# OpenReview forum: "Revisiting Adversarial Robustness Distillation from the Perspective of Robust Fairness"
_NeurIPS.cc/2023/Conference — NeurIPS 2023 poster_

### Official Review · Reviewer_3Z9v · 2023-07-05

**Soundness:** 3 good
**Presentation:** 3 good
**Contribution:** 3 good
**Rating:** 6
**Confidence:** 2

**Summary:**

The paper addresses the issue of amplified class-imbalanced performance during adversarial robust distillation (ARD). The authors propose a fair-oriented ARD method that incorporates a class re-weighting mechanism for adversarial training. The method is applicable to various ARD techniques and consistently achieves success across different datasets.


**Strengths:**

1. The paper provides strong evidence (Fig. 1 and Tab. 1) to highlight the severity of the robust fairness (class-imbalanced adversarial robustness) issue.
2. The proposed method is simple yet effective, as validated through extensive experiments.


**Weaknesses:**

1. The reviewer suggests investigating whether existing fair-oriented distillation methods can mitigate the robust fairness problem with slight modifications. For example, [A] addressed biased issues in standard knowledge distillation and conducted experiments on the CIFAR-10S dataset. It would be beneficial to discuss and compare these existing methods in this submission.

2. Prior work [17, 29] has emphasized that average accuracy can be odd to fairness in both standard and adversarial robust settings. However, the experiments presented in Tab. 2, Tab. 4 of the main paper, as well as Tab. 7 and Tab. 8 in the supplementary material, show that the proposed method consistently achieves significantly better average scores while maintaining the best fairness scores, contrary to established theories. This raises concerns about the credibility of the experiments, and the reviewer requests an explanation for this phenomenon.

[A] Jung, Sangwon, et al. "Fair feature distillation for visual recognition." Proceedings of the IEEE/CVF conference on computer vision and pattern recognition. 2021.



**Questions:**

1. The term "fairness" has been overloaded in recent years. The authors seem to refer to the "class-imbalance" issue and evaluate it using "minimax fairness." However, it is important to note that conventional fairness problems typically involve specified sensitive groups, which are orthogonal to predicted classes, and require multiple evaluation metrics such as demographic parity and equalized odds. It is recommended that the authors address more fairness topics in the paper, perhaps in the related work section, to provide clarity.

2. The authors appear to have missed relevant related work, such as [B], which discusses fairness through robustness.

[B] Nanda, Vedant, et al. "Fairness through robustness: Investigating robustness disparity in deep learning." Proceedings of the 2021 ACM Conference on Fairness, Accountability, and Transparency. 2021.


**Limitations:**

Yes.

---

> ### Author Rebuttal · Authors · 2023-08-09
>
> Thank you for your constructive comments. We have taken great care to address all your concerns, providing necessary clarifications when needed. We genuinely hope that our responses comprehensively answer your questions.
>
> > **(Comment 1): The reviewer suggests investigating whether existing fair-oriented distillation methods can mitigate the robust fairness problem with slight modifications. For example, [A] addressed biased issues in standard knowledge distillation and conducted experiments on the CIFAR-10S dataset.**
>
> **Answer 1:**
>
> **Firstly**, we will clarify the difference between fairness in existing fairness-oriented distillation methods and the fairness we are addressing.
>
> 1) Their meanings differ. Prior work on fair distillation [A] aims to prevent discriminatory biases stemming from sensitive attributes. Our robust fairness focuses instead on minimizing differences in robustness across classes.
>
> 2) Their metrics for measuring fairness differ. [A] measures fairness via equalized odds metrics $DEO_M$ and $DEO_A$ which account for sensitive attributes. In contrast, our metrics of the worst-class accuracy and NSD are solely based on model performance across classes.
>
> **Secondly**, we thoroughly investigated existing fair distillation methods to assess modifiability for addressing robust fairness. [A] proposes MMD-based Fair Distillation (MFD) to mitigate algorithmic bias via feature distillation. Unfortunately, without available open-source code, reproducing [A] was infeasible within the given timeframe. Nevertheless, we will include a detailed discussion of [A] and experimental results on adapted MFD for robust fairness improvements in the revision of our paper.
>
> > **(Comment 2): Prior work [17, 29] has emphasized that average accuracy can be odd to fairness in both standard and adversarial robust settings. However, the experiments presented in Tab. 2, Tab. 4 of the main paper, as well as Tab. 7 and Tab. 8 in the supplementary material, show that the proposed method consistently achieves significantly better average scores while maintaining the best fairness scores, contrary to established theories. This raises concerns about the credibility of the experiments, and the reviewer requests an explanation for this phenomenon.**
>
> **Answer 2:**
>
> Although prior work [17, 29] shows such a tradeoff between accuracy and fairness, we consider that the conclusions from these two studies may lack generalizability due to differences in settings compared to our work.
>
> **Firstly**, FRL [17] improves fairness but reduces average accuracy. However, [17] does not theoretically analyze whether average accuracy inherently causes fairness biases in standard and robust settings. Notably, CFA [C] improves both accuracy and fairness over FRL, indicating that the accuracy-fairness tradeoff may stem from the design of FRL rather than the relationships between accuracy and fairness.
>
> **Secondly**, [29] theoretically and empirically analyzed the robustness-fairness tradeoff. However, as shown in Tables 3 and 4 in [29], their FAT method attains superior robustness and robust fairness over Madry (SAT [18]), contradicting the proposed trade-off relationship. This discrepancy may arise from two factors: 1) Their theoretical analyses use simplified linear models, differing from large-scale deep neural networks. Thus conclusions may not extend. 2) The motivational experiments in [29] use inconsistent perturbation radii between training and testing, with training perturbation radii smaller than testing perturbation radii. This risks insufficient robustness gains, potentially restricting conclusion generalizability to standard equal perturbation settings. Therefore, the conclusion of [29] may not generalize to general settings.
>
> [C] Wei, Zeming et al. CFA: Class-wise calibrated fair adversarial training. In *CVPR*, 2023.
>
> > **(Comment 3): The term "fairness" has been overloaded in recent years. The authors seem to refer to the "class-imbalance" issue and evaluate it using "minimax fairness." However, it is important to note that conventional fairness problems typically involve specified sensitive groups, which are orthogonal to predicted classes, and require multiple evaluation metrics such as demographic parity and equalized odds. It is recommended that the authors address more fairness topics in the paper, perhaps in the related work section, to provide clarity.**
>
> **Answer 3:**
>
> Thank you for your valuable suggestions. We will add a detailed discussion on fairness topics in the related work section.
>
> **Firstly,** we clarify the distinction between our notion of robust fairness and conventional machine learning (ML) fairness. The robust fairness pertains to consistent predictive accuracy across classes under both standard and adversarial conditions [17]. In contrast, conventional ML fairness involves eliminating bias based on sensitive attributes like gender or race.
>
> **Secondly,** prior works have studied fairness in classification, e.g., equalized odds [D]. However, to the best of our knowledge, we are the first to investigate the problem of achieving adversarially robust fairness through knowledge distillation.
>
> [D] Hardt, Moritz et al. Equality of opportunity in supervised learning. In *NeurIPS*, 2016.
>
> > **(Comment 4): The authors appear to have missed relevant related work, such as [B], which discusses fairness through robustness.**
>
> **Answer 4:**
>
> Thank you for your suggestion. We will discuss more related works, including [B], in the revision of our paper.
>
> [B] introduced the concept of robustness bias, pointing out that different classes of data should have the same robustness, but did not propose specific implementation methods. [B] also did not consider the impact of knowledge distillation on robust fairness. Furthermore, the fairness metrics used in [B] differ from ours. [B] uses their self-defined AUC, while we utilize the worst-class robustness and NSD.

---

### Official Review · Reviewer_5xrn · 2023-07-06

**Soundness:** 2 fair
**Presentation:** 2 fair
**Contribution:** 2 fair
**Rating:** 5
**Confidence:** 3

**Summary:**

This paper focuses on the issue of robust fairness in adversarial robustness distillation. This paper proposes the Fair Adversarial Robustness Distillation (Fair-ARD) framework, which utilizes a more refined difficulty metric and an adaptive class re-weighting approach, enabling the student model to learn the knowledge about each class in a fairer way. Extensive experiments confirm that Fair-ARD can significantly improve the robust fairness of the student model and slightly improve overall robustness.

**Strengths:**

+The main strength paper firstly investigates the issue of robust fairness in adversarial robustness distillation, and furthermore proposed a reweighting method to get a class-fair student model.

+The paper is well-writing and easy to follow.


**Weaknesses:**

-The novelty is somewhat limited.  Robust fairness is widely studied in the adversarial training method (adversarial distillation is a type of variant to improve the robustness based on the adversarial training). In this regard, the originality of the research is limited.

-The reweighting method toward different class is widely applied in adversarial training ([1], [2] as mentioned in article). More advanced state-of-the-art method (such as [3]) should be discussed and compared.

-The metric to measure class difficulty is lack of theoretical basis. Directly applying the least PGD steps seem to not be a good estimate method. Different step size and steps num will lead to different results, e.g., Auto-PGD [4], and the similar idea is applied in AT (e.g. [2], [5]). If the least PGD steps is applied in reweighting and can really be helpful, a theoretical analysis is supposed to provide.

[1] Jingfeng Zhang, Jianing Zhu, Gang Niu, Bo Han, Masashi Sugiyama, and Mohan Kankanhalli. 360 Geometry-aware instance-reweighted adversarial training. In ICLR, 2021.

[2] Han Xu, Xiaorui Liu, Yaxin Li, Anil Jain, and Jiliang Tang. To be robust or to be fair: Towards 355 fairness in adversarial training. In ICML, 2021.

[3] Wei, Zeming, et al. "Cfa: Class-wise calibrated fair adversarial training." Proceedings of the IEEE/CVF Conference on Computer Vision and Pattern Recognition. 2023.

[4] Francesco Croce and Matthias Hein. Reliable evaluation of adversarial robustness with an  ensemble of diverse parameter-free attacks. In International conference on machine learning, 362 pages 2206–2216. PMLR, 2020.

[5] Xiaojun Jia, Yong Zhang, Baoyuan Wu, Ke Ma, Jue Wang, Xiaochun Cao; Proceedings of the IEEE/CVF Conference on Computer Vision and Pattern Recognition (CVPR), 2022.

**Questions:**

see weakness

**Limitations:**

see weakness

---

> ### Author Rebuttal · Authors · 2023-08-09
>
> Thank you for your constructive comments. We have taken great care to address all your concerns, providing necessary clarifications when needed. We genuinely hope that our responses comprehensively answer your questions.
>
> >**(Comment 1): The novelty is somewhat limited ...**
>
> **Answer 1:**
>
> To begin with, you are correct that adversarial robust distillation (ARD) is an improvement based on adversarial training (AT). However, compared to existing works about AT, our novelty and originality are reflected in the following three aspects:
>
> **Firstly**, although ARD is an improved method based on AT, it significantly differs from AT. Such differences warrant special consideration for ARD.
>
> 1) Their training processes are different. ARD achieves robustness by allowing the student model to mimic the teacher model's representations on adversarial examples, thereby transferring the teacher model's robustness to the student model [6]. In contrast, AT involves adversarial examples in the training process to enhance the model's robustness [7].
>
> 2) Their goals are different. ARD aims to improve the adversarial robustness of small models (e.g., ResNet18) for resource-constrained scenarios [6], while AT frequently uses relatively larger models (e.g., WRN34-10) [8]. And model capacity is a critical factor for robustness [1].
>
> 3) Their considerations about robust fairness are different. For ARD, both the teacher model's robust fairness and the influence of distillation on the student's inherited fairness should be considered. In contrast, AT only needs to examine how the training process impacts the robust fairness of the model itself.
>
> **Secondly**, the robust fairness issue is important but **has not been studied in ARD**, and the terrible robust fairness of ARD may significantly hinder its application. To the best of our knowledge, **we are the first to investigate the issue of robust fairness in ARD**. We reveal that the student model can only partially inherit the teacher model's robust fairness and provide insights into the internal reasons.
>
> **Finally**, existing methods for improving robust fairness in AT do not perform as well when applied to ARD. In the discussion section of our paper, we have attempted to apply existing fair AT algorithms, such as FRL [2] and FAT [9], to ARD. Experimental results demonstrate that our Fair-ARD significantly outperforms these fair algorithms.
>
> Therefore, we believe that our work exhibits substantial novel contributions compared to prior works.
>
> [6] Goldblum, Micah et al. Adversarially robust distillation. In *AAAI*, 2020.
>
> [7] Aleksander Madry et al. Towards deep learning models resistant to adversarial attacks. In *ICLR*, 2018.
>
> [8] Boxi Wu et al. Do wider neural networks really help adversarial robustness? In *NeurIPS*, 2021.
>
> [9] Xinsong Ma et al. On the tradeoff between robustness and fairness. In *NeurIPS*, 2022.
>
> > **(Comment 2): More advanced state-of-the-art method (such as [3]) should be discussed and compared.**
>
> **Answer 2:**
>
> We compare our method and CFA [3] across two dimensions: design and experimental performance.
>
> **Firstly**, our method diverges from CFA in three fundamental aspects:
>
> 1) Different metrics for measuring class difficulty: CFA uses the model's robustness on each class as the difficulty metric, aligning with FRL [2] Example Difficulty (FED). Figure 2 of our paper depicts that our example difficulty metric LPS furnishes finer granularity than FED. So, our class difficulty measure $\kappa$ provides more fine-grained estimates than the class-wise robustness used by CFA.
>
> 2) Different reweighting functions: CFA applies a fractional function based on class difficulty, whereas we primarily use power functions for reweighting, and we discuss various types of reweighting functions in Appendix C.4. Based on the experimental results shown in the table below, it appears that our reweighting function might be more suitable.
>
> 3) CFA utilizes exponential moving average (EMA), while Fair-ARD does not employ it.
>
> **Secondly**, we have compared Fair-ARD against CFA by implementing its method in ARD (CFA-ARD). As shown in the partial table below and the detailed Table 16 in the global rebuttal PDF, CFA has higher robust fairness than ARD, but CFA-ARD has lower robust fairness than Fair-ARD. This highlights Fair-ARD's strength in enhancing robust fairness.
>
> |Method|Clean Avg.|Clean Worst|AA Avg.|AA Worst|
> |:-:|:-:|:-:|:-:|:-:|
> |ARD|**83.22**|61.00|49.05|16.90|
> |CFA-ARD|82.00|62.30|**49.73**|20.00|
> |Fair-ARD (ours)|82.96|**68.10**|49.13|**29.20**|
>
> > **(Comment 3): The metric to measure class difficulty is lack of theoretical basis ...**
>
> **Answer 3:**
>
> Although our use of LPS (least PGD steps) lacks theoretical basis, empirical results demonstrate its efficacy. Firstly, GAIRAT [1] validated LPS for estimating difficulty. The left panel of Figure 3 shows more attackable examples with smaller LPS are closer to the decision boundary. Similarly, the right panel displays examples with smaller LPS values nearer the boundary. These trends affirm LPS as an effective difficulty metric. Furthermore, our extensive experiments exhibit Fair-ARD's robust fairness improvements over ARD, corroborating LPS's utility. Secondly, the PGD step size and number are fixed when computing LPS during adversarial example generation in our distillation process. This eliminates potential variations from differing step configurations, stabilizing the results.

---

> > ### Comment · Reviewer_5xrn · 2023-08-17
> >
> > Thank you for your response. My issues have been addressed.

---

### Official Review · Reviewer_DXyp · 2023-07-06

**Soundness:** 3 good
**Presentation:** 3 good
**Contribution:** 3 good
**Rating:** 7
**Confidence:** 4

**Summary:**

Adversarial robustness distillation transfers knowledge from a teacher model to a student model and improves the overall robustness of the student in terms of resisting adversarial attacks. This work studies robust fairness in the task of adversarial robustness distillation, which focuses on the adversarial robustness of different classes. This work first makes an observation that the student may only partially inherit the teacher's robust fairness, due to the capacity gap between the teacher and student, and the varying class difficulties. To address this issue of robust fairness, this work proposes Fair-ARD, which uses a geometric perspective to quantify the per-class difficulty and adjust the weight of each class accordingly. Extensive experiments demonstrate the efficacy of Fair-ARD in improving the robust fairness in knowledge distillation.

**Strengths:**

1. This work, for the first time, explores how to transfer robust fairness in adversarial robustness distillation. Robust fairness can be of vital importance in ML applications with social and/or safety concerns. This research initiates an important direction for the community.
2. The proposed method, Fair-ARD, is a general framework that can be readily incorporated into existing ARD methods.
3. The experiment evaluation is thorough.
4. The writing and math notations are clear.

**Weaknesses:**

1. The experiments are mostly done on CIFAR-10/100, a relatively curated and easy dataset. Following prior work like IAD [35], it is encouraged to include more realistic and challenging datasets like Tiny-ImageNet. In the sense of robust fairness, a large number of classes and/or unbalanced data distribution would introduce some more practical challenges.
2. The re-weighting strategy in proposed method requires re-estimating the difficulty of every sample in every epoch. This process involves multiple PGD steps. This method may introduce a large quantity of computation overhead to the distillation procedure.
3. Citations should be given when previous methods first appear in the paper (e.g., IAD, RSLAD, and MTARD in Line 75).

**Questions:**

1. In Figure 2, the 10-th adversarial example of $\tilde x_4$ is still correctly classified. Why the LPS value of $x_4$ is 10? This seems to contradict with Equation (5).
2. Also in Equation (5), how to determine $d(x,y)$ if the adversarial examples within the $\epsilon$-ball never move across the decision boundary?

**Limitations:**

Please check 1. in Weaknesses.

---

> ### Author Rebuttal · Authors · 2023-08-09
>
> Thank you for your constructive comments. We have taken great care to address all your concerns, providing necessary clarifications when needed. We genuinely hope that our responses comprehensively answer your questions.
>
> > **(Comment 1): The experiments are mostly done on CIFAR-10/100, a relatively curated and easy dataset. Following prior work like IAD [35], it is encouraged to include more realistic and challenging datasets like Tiny-ImageNet. In the sense of robust fairness, a large number of classes and/or unbalanced data distribution would introduce some more practical challenges.**
>
> **Answer 1:**
>
> We have conducted experiments on the imbalanced SVHN [1] and large-scale Tiny-ImageNet [2] datasets. Using ResNet18 and IAD [3] as an example, our Fair-ARD approach achieved improved worst-class robustness against AA attacks, with enhancements of **3.98%** on SVHN and **0.45%** on Tiny-ImageNet compared to IAD. These consistent results further validate the effectiveness and generalization of our Fair-ARD.
>
> |    Dataset    |     Method      | Clean Avg. | Clean Worst |  AA Avg.  | AA Worst/Worst-20% |
> | :-----------: | :-------------: | :--------: | :---------: | :-------: | :----------------: |
> |     SVHN      |       IAD       |   90.28    |    74.04    |   43.58   |       20.12        |
> |     SVHN      | Fair-IAD (ours) | **91.66**  |  **83.19**  | **44.55** |     **24.10**      |
> | Tiny-ImageNet |       IAD       |   40.06    |    15.95    |   14.46   |        1.15        |
> | Tiny-ImageNet | Fair-IAD (ours) | **40.94**  |  **16.80**  | **15.35** |      **1.60**      |
>
> > **(Comment 2): The re-weighting strategy in proposed method requires re-estimating the difficulty of every sample in every epoch. This process involves multiple PGD steps. This method may introduce a large quantity of computation overhead to the distillation procedure.**
>
> **Answer 2:**
>
> Apologies for the misunderstanding - to clarify, estimation of difficulty does not require any additional PGD steps to the distillation procedure.
>
> During the training process, ARD itself requires generating adversarial examples to improve robustness. Our method just leverages this process to estimate difficulty, without introducing any additional PGD steps. As a result, our Fair-ARD only adds negligible computation overhead compared to the original PGD operations. For specific computational overhead on training time and GPU memory, please refer to Appendix C.7 and Table 13 of our paper. Table 13 shows our method slightly increases computation time but does not increase GPU memory usage.
>
> |      Method       | Time (Avg. Epoch) | GPU Memory |
> | :---------------: | :---------------: | :--------: |
> |        ARD        |      130.14s      |  3645MiB   |
> |  Fair-ARD (ours)  |      154.96s      |  3645MiB   |
> |        IAD        |      167.22s      |  4231MiB   |
> |  Fair-IAD (ours)  |      207.70s      |  4231MiB   |
> |       RSLAD       |      161.50s      |  3877MiB   |
> | Fair-RSLAD (ours) |      215.23s      |  3877MiB   |
> |       MTARD       |      229.96s      |  4017MiB   |
> | Fair-MTARD (ours) |      232.87s      |  4017MiB   |
>
> > **(Comment 3): Citations should be given when previous methods first appear in the paper (e.g., IAD, RSLAD, and MTARD in Line 75).**
>
> **Answer 3:**
>
> Thank you for your valuable suggestion. We will add these details in the revised version.
>
> > **(Comment 4): In Figure 2, the 10-th adversarial example of $\tilde{x}_4$ is still correctly classified. Why the LPS value of $x_4$ is 10? This seems to contradict with Equation (5).**
>
> **Answer 4:**
>
> Following GAIRAT [4], the LPS value is constrained to the range $[0,K]$, where $K$ represents the maximum number of steps for generating adversarial examples during training. Since $K$ is set to 10 for the adversarial robustness distillation methods discussed in this paper (Appendix C.1, line 452), the LPS of example $x_4$ will be 10 even if its 10th adversarial example is correctly classified.
>
> This is a special case for Equation 5. We will introduce a constraint, namely $d(x,y)\in[0,10]$, to ensure consistency between Equation (5) and the examples shown in Figure 2.
>
> > **(Comment 5):  Also in Equation (5), how to determine $d(x,y)$ if the adversarial examples within the $\epsilon$-ball never move across the decision boundary?**
>
> **Answer 5:**
>
> As illustrated in Answer 4,  $d(x,y)$ is $K$ in this case. Since $K$ is 10 in the adversarial robustness distillation methods discussed in our paper, $d(x,y)$ is actually 10 if the adversarial examples within the $\epsilon$-ball never move across the decision boundary.
>
> Our computation of $d(x,y)$ is as follows:
>
> 1. $\tilde{x} \leftarrow x ; d(x, y) \leftarrow 0$
> 2. $\textbf{while }K>0\textbf{ do}$
> 3. $\quad \textbf{if } \arg\max_i f(\tilde{x})=y \textbf{ then}$
> 4. $\quad\quad d(x, y) \leftarrow d(x, y)+1$
> 5. $\quad \textbf{end if }$
> 6. $\quad \tilde{x} \leftarrow \Pi_{\mathcal{B}}[x, \epsilon]\left(\gamma \operatorname{sign}\left(\nabla_{\tilde{x}} \ell(f(\tilde{x}), y)\right)+\tilde{x}\right)$
> 7. $\quad K \leftarrow K-1$
> 8. $\textbf{end while }$
>
> Hence, if an adversarial example within the $\epsilon$-ball never crosses the decision boundary, meaning the condition in line 3 of the code above is always true, $d(x,y)$ will accumulate $K$ times, and the value of $d(x,y)$ will be $K$, i.e., 10.
>
> [1] Yuval Netzer et al. Reading digits in natural images with unsupervised feature learning. 2011.
>
> [2] Ya Le et al. Tiny imagenet visual recognition challenge. In *CS 231N*, 2015.
>
> [3] Wei, Zeming et al. Cfa: Class-wise calibrated fair adversarial training. In *CVPR*, 2023.
>
> [4] Jingfeng Zhang et al. Geometry-aware instance-reweighted adversarial training. In *ICLR*, 2021.

---

> > ### Comment · Reviewer_DXyp · 2023-08-17
> >
> > Thank you for your response. I have read your answers to my questions and the other reviews. My concerns have been adequately addressed.

---

### Official Review · Reviewer_Mqsm · 2023-07-08

**Soundness:** 3 good
**Presentation:** 3 good
**Contribution:** 2 fair
**Rating:** 6
**Confidence:** 4

**Summary:**

This paper deals with the issue of fairness in adversarial training in image classification with distillation. The idea is that due to model capacity gap between the teacher and the student model, not all classes are equally capable of retaining the teacher model's robustness and thus the distilled student model shows a larger variation among different classes in terms of adversarial robustness which is not desired.
The proposed solution is a simple class-wise re-weighting of the training objective which increases the importance of such difficult classes and reduces that of the easier classes. The proposed weights are heuristically motivated according to the (inverse) of the number of PGD steps needed to find an adversarial example averaged over each class (weighted with hyper-parameter beta).

Numerical results on CIFAR 10 and CIFAR100 show empirical improvements of the average and worst-class robustness.

**Strengths:**

The problem is interesting and well-motivated. The paper is written well and easy to follow. The proposed solution is intuitive and simple.

Empirical results show improvement in terms of average and worst-class adversarial robustness against state-of-the-art on cifar 10 and cifar 100.

**Weaknesses:**

The paper proposed a simple re-weighting approach, and lacks any foundational contribution to the problem. The performance is thus heavily judged by empirical tests.

My concern on this issue is the following:

1- Due to lack of theoretical grounding, how reliable are the experimental results? This is specially important in CIFAR100 results given the low robustness (<5%). One could for instance report the standard deviation across different training runs to show such sensitivity.

2- The formulation is heuristic, especially as the proposed class weights are not predetermined and change according to the latest model obtained in the last iteration. This means the objective changes in each epoch. How stable is this approach? Also, what are these weights calculated over (training set or validation set)? As one of the differences with GAIRAT it is mentioned that least PGD steps (LPS) is calculated over the validation set in there. How important is this choice and how much difference does it make?

3- The metric to measure class difficulty is borrowed from GAIRAT, with some modifications. What is the sensitivity to the pgd parameters such as the step-size.

**Questions:**

1- Given the empirical nature of the results, how sensitive is the model versus the PGD step-size in LPS? How would the results change if $gamma$ was 10 times smaller for instance?

2- As motivating examples, it is mentioned that the gap between student and teacher is the reason behind worse fairness. However, tests show (see Fig1 for instance) that teacher models performance are often surpasses by the student model once the student model size reaches that of the teacher. This means that the size of the model is only partly responsible for the performance deterioration and contradicts the statement in my opinion.

3- In the abstract two relative improvements are mentioned (12.3% and 5.3%) while both refer to CIFAR10 with Resnet18. Could you please clarify.

**Limitations:**

Yes

---

> ### Author Rebuttal · Authors · 2023-08-09
>
> Thank you for your constructive comments. We have taken great care to address all your concerns, providing necessary clarifications when needed. We genuinely hope that our responses comprehensively answer your questions.
>
> > **(Comment 1): Due to lack of theoretical grounding, how reliable are the experimental results?  ...**
>
> **Answer 1:**
>
> Thank you for your valuable suggestion. All training runs were independently conducted five times and reported averages, omitting negligible standard deviations (illustrated in line 198 of our paper). We have presented the results of using ResNet18 on CIFAR100 in the partial table below and the detailed Table 14 in the global rebuttal PDF. These minor standard deviations align with prior AT work [1,2], further demonstrating the stability. We will include the standard deviations for each experiment in the revision to provide full transparency and rigor regarding our experimental process.
>
> |Method|Clean Avg.|Clean Worst|AA Avg.|AA Worst|
> |:-:|:-:|:-:|:-:|:-:|
> |Fair-IAD (ours)|57.08±0.34|18.00±0.33|25.55±0.26|2.00±0.65|
> |Fair-RSLAD (ours)|58.33±0.45|21.00±0.48|26.37±0.51|3.00±0.51|
> |Fair-MTARD (ours)|57.72±0.21|22.00±0.41|26.18±0.53|3.00±0.43|
>
> [1] Chen Liu et al. On the loss landscape of adversarial training: Identifying challenges and how to overcome them. In *NeurIPS*, 2020.
>
> [2] Boxi Wu et al. Do wider neural networks really help adversarial robustness? In *NeurIPS*, 2021.
>
> > **(Comment 2): The formulation is heuristic, especially as the proposed class weights are not predetermined and change ...**
>
> **Answer 2:**
>
> 1)  Stability of the Method
>
> Figure 10 in the global rebuttal PDF presented the variations in the weights of each class during training.  And we found that the weights initially fluctuated but gradually stabilized in the later stages.
>
> 2)  Weight Computation
>
> Both GAIRAT and our method calculate weights using the training set but differ in the subsets utilized for each calculation. We calculate weights across the entire training set, updating them once per epoch. In contrast, GAIRAT calculates weights on each mini-batch and updates them after every mini-batch. Since each batch may have different class distributions, this can potentially influence the measurement of class difficulty. Hence, our method can obtain a more accurate assessment by determining the class difficulty over the entire training set. The results in Tables 4 and 12 demonstrate our method's superiority.
>
> > **(Comment 3): The metric to measure class difficulty is borrowed from GAIRAT, with some modifications. What is the sensitivity to the pgd parameters such as the step-size.**
>
> **Answer 3:**
>
> We analyze the sensitivity of the step size theoretically and empirically as follows.
>
> Increasing step size $\gamma$ reduces steps for examples to cross decision boundaries, diminishing LPS discrimination. As shown in the partial table below and the detailed Table 15 in the global rebuttal PDF, $\gamma$=3/255 versus 2/255 maintains average robustness but worsens worst-class, due to weakened re-weighting from reduced LPS discrimination. However, Fair-ARD still exceeds ARD [3].
>
> Conversely, decreasing γ can raise steps to boundaries, potentially improving discrimination. But smaller $\gamma$ risks lower quality adversarial examples for training, hindering distillation. As shown in the partial table below and the detailed Table 15 in the global rebuttal PDF, a 10x smaller $\gamma$ substantially decreased robustness, which can be attributed to poorer adversarial examples.
>
> In summary, $\gamma$ tuning balances discrimination versus adversarial training efficacy. Careful adjustment optimizes Fair-ARD performance.
>
> |Method|Step Size|Clean Avg.|Clean Worst|AA Avg.|AA Worst|
> |:-:|:-:|:-:|:-:|:-:|:-:|
> |ARD|2/255|83.22|61.00|49.05|16.90|
> |Fair-ARD (ours)|0.2/255|**86.22**|**70.30**|36.39|14.90|
> |Fair-ARD (ours)|1/255|84.35|69.90|47.95|22.80|
> |Fair-ARD (ours)|2/255|82.96|68.10|**49.13**|**29.20**|
> |Fair-ARD (ours)|3/255|83.90|69.60|49.10|27.00|
>
> > **(Comment 4): Given the empirical nature of the results, how sensitive is the model versus the PGD step-size in LPS? ...**
>
> **Answer 4:**
>
> Please refer to Answer 3 for detailed analyses.
>
> > **(Comment 5): As motivating examples, it is mentioned that the gap between student and teacher is the reason behind worse fairness ...**
>
> **Answer 5:**
>
> We first clarify that our conclusion aligns with yours - the capacity gap between teacher and student models is the main reason (Line 45 in our paper) rather than the sole reason for the robust fairness issues. In the following, we illustrate the capacity gap as a main reason because:
>
> **Firstly,** as shown in (b)~(d) of Fig1, continuously increasing the student model capacity leads to consistent improvements in robust fairness across larger model sizes. This clear trend corroborates our proposition that model capacity is a key factor influencing robust fairness.
>
> **Secondly,** since distillation itself can enhance model performance [3], if a pre-trained model is distilled using itself as the teacher, its performance will be boosted even more. Consequently, when model architectures are the same, distillation can heighten robust fairness, harmoniously aligning with our perspective.
>
> [3] Goldblum, Micah et al. Adversarially robust distillation. In *AAAI*, 2020.
>
> > **(Comment 6): In the abstract two relative improvements are mentioned (12.3% and 5.3%) while both refer to CIFAR10 with Resnet18. Could you please clarify.**
>
> **Answer 6:**
>
> 1)  12.3%:
>
> On ResNet18 and CIFAR10, our Fair-ARD improved the worst-class robustness under AA by 12.3% compared to vanilla ARD.
>
> 2)  5.3%:
>
> On ResNet18 and CIFAR10, our Fair-ARD improved the worst-class robustness under AA by 5.3% compared to FRL-ARD, which refers to applying FRL [4] techniques to vanilla ARD.
>
> [4] Xu, Han et al. To be robust or to be fair: Towards fairness in adversarial training. In *ICML*, 2021.

---

> > ### Comment · Reviewer_Mqsm · 2023-08-21
> > **Response**
> >
> > I would like to thank the authors for their response. My concerns are mostly answered, and I have raised my rating to 6.

---

### Official Review · Reviewer_qqxE · 2023-07-10

**Soundness:** 4 excellent
**Presentation:** 4 excellent
**Contribution:** 4 excellent
**Rating:** 7
**Confidence:** 4

**Summary:**

This work studies the problem of fairness in the context of distillation of adversarially robust models. Specifically, it is shown that the accuracy disparity between classes (leading to “unfairness”) is _further exacerbated_ for student models, attributed to the a) capacity gap between the student and teacher model and b) equal importance given to classes during the distillation process.
This essentially leads to the need for fair adversarial distillation. This is achieved via modelling the “difficulty of learning a class” via the Least PDG Steps (LPS) measure to model the ease by which a sample can cross the decision boundary. Based on this, a flexible and general reweighting scheme is proposed which can be applied “post-hoc” on top of any distillation mechanisms by allowing for the reweighting. A thorough empirical evaluation is presented showing consistent improvements in the average and worst-group accuracy over existing methods as well as superiority over other fair adversarial training methods.

**Strengths:**

This paper studies an important and relevant problem of the fairness-robustness trade-off in the relevant setting of knowledge distillation. I like the motivation behind the need to study the "fine-grained" effect of adversarial distillation w.r.t accuracy gap between different classes. The approach to analyse the problem is concise and well motivated: and has important implications in both the "inductive bias" of adversarial robustness towards different classes as well as the generality of reweighting via class difficulty. Further, the empirical evaluation is thorough and comparisons are made with comprehensive baselines, both w.r.t original robust-distillation methods as well as different "reweighting" / fairness methods.



**Weaknesses:**

Following are some questions which I believe should be addressed:
1. An essential baseline assumption is that _all classes are balanced_ i.e. equally represented in the training data. How do you foresee the generality of your method in the scenario where we not only consider class difficulty but also the class-imbalance? It looks like the estimator on Eq(6) would now be biased.
2. The results seem to be evaluated only on the CIFAR datasets, it would be interesting to see how this observation holds beyond CIFAR.


**Questions:**

Please see section on Weakenesses.

**Limitations:**

While discussion on the choice of $\beta$ is presented, it would be nice if any limitations of the existing method (such as reweighting) are discussed.

---

> ### Author Rebuttal · Authors · 2023-08-09
>
> Thank you for your constructive comments. We have taken great care to address all your concerns, providing necessary clarifications when needed. We genuinely hope that our responses comprehensively answer your questions.
>
> > **(Comment 1): An essential baseline assumption is that all classes are balanced i.e. equally represented in the training data. How do you foresee the generality of your method in the scenario where we not only consider class difficulty but also the class-imbalance? It looks like the estimator on Eq(6) would now be biased.**
>
> **Answer 1:**
>
> Our Fair-ARD can also generalize to the scenario where both class difficulty and imbalance exist.
>
> **Firstly**, although the datasets used in this paper are balanced across classes, the number of examples in each class has a relatively minor impact on the $\kappa_i$ metric, since it is calculated by averaging the difficulty of all examples within the $i$-th class as shown in Eq. (6).
>
> **Secondly**, to address your question about simultaneously considering class difficulty and class imbalance, we have conducted experiments on the SVHN dataset [1], which is an imbalanced dataset with different numbers of training examples across classes.
> Using ResNet18 and IAD [2] as an example, our Fair-ARD framework achieved an enhanced worst-class robustness of **24.10%** against AA attacks, **3.98%** higher than IAD. This positive result on an imbalanced dataset further validates the generality of our method in tackling the combined challenges of class difficulty and imbalance. We will include these supportive experimental results in the revision of the paper.
>
>
> > **(Comment 2): The results seem to be evaluated only on the CIFAR datasets, it would be interesting to see how this observation holds beyond CIFAR.**
>
> **Answer 2:**
>
> Our observation also holds beyond CIFAR datasets. We have validated our viewpoint on the SVHN and Tiny-ImageNet [3] datasets on ResNet18. Since the worst-class robustness is extremely low and there are only 50 images for each class in the test set, we report the average of the worst-20% class robustness on Tiny-ImageNet following [4]. As the table below shows, compared to IAD, our Fair-ARD framework improved worst-class (worst-20% class) robustness under AA by **3.98%** (**0.45%**) on SVHN (Tiny-ImageNet). These consistent enhancements on additional benchmark datasets further demonstrate the generality of our method.
>
>
> |    Dataset    |     Method      | Clean Avg. | Clean Worst |  AA Avg.  | AA Worst/Worst-20% |
> | :-----------: | :-------------: | :--------: | :---------: | :-------: | :----------------: |
> |     SVHN      |       IAD       |   90.28    |    74.04    |   43.58   |       20.12        |
> |     SVHN      | Fair-IAD (ours) | **91.66**  |  **83.19**  | **44.55** |     **24.10**      |
> | Tiny-ImageNet |       IAD       |   40.06    |    15.95    |   14.46   |        1.15        |
> | Tiny-ImageNet | Fair-IAD (ours) | **40.94**  |  **16.80**  | **15.35** |      **1.60**      |
>
> > **(Comment 3): While discussion on the choice of $\beta$ is presented, it would be nice if any limitations of the existing method (such as reweighting) are discussed.**
>
> **Answer 3:**
>
> Thank you for your valuable suggestion. We will briefly discuss the limitations of our method here and include a detailed discussion in the revision of our paper.
>
> Regarding the limitations of our method, the reweighting function is heuristic and may not be optimal. Given the diverse perspectives in adversarial training, there may exist more effective methods. Besides, our adopted reweighing algorithm is an intuitive design. More fine-grained reweighting methods may achieve higher worst-class robustness.
>
> In addition, our work primarily focuses on discovering and alleviating the robust fairness issue in Adversarial Robustness Distillation. We leave the exploration of designing more effective solutions for future work.
>
> [1] Yuval Netzer et al. Reading digits in natural images with unsupervised feature learning. 2011.
>
> [2] Jianing Zhu et al. Reliable adversarial distillation with unreliable teachers. In *ICLR*, 2021.
>
> [3] Ya Le et al. Tiny imagenet visual recognition challenge. In *CS 231N*, 2015.
>
> [4] Wei, Zeming et al. Cfa: Class-wise calibrated fair adversarial training. In *CVPR*, 2023.

---

> > ### Comment · Reviewer_qqxE · 2023-08-20
> >
> > Thank you Authors for clarifying, I acknowledge that I have read the rebuttal.

---

### Author Rebuttal · Authors · 2023-08-10

Figure 10: The weights of each class in Fair-ARD using ResNet18 on CIFAR-10 across epochs.

Table 14: The average (Avg.) and worst-class (Worst) robustness and their standard deviations for
various algorithms using ResNet18 on CIFAR-100.

Table 15: The average (Avg.) and worst-class (Worst) robustness for various algorithms and step
sizes using ResNet18 on CIFAR-10. The best results are **bolded**, and the second best results are
underlined.

Table 16: The average (Avg.) and worst-class (Worst) robustness for various algorithms using
ResNet18 on CIFAR-10. The best results are **bolded**.

---

### Comment · Area_Chair_9etX · 2023-08-18
**Kindly reminder to respond to author responses**

Dear Reviewers,

Thank you very much again for performing this extremely valuable service to the NeurIPS authors and organizers.

As the authors have provided detailed responses, it would be great if you could check them and see if your concerns have been addressed. Given that the discussion phase between authors and reviewers is approaching the end in the next few days, your prompt feedback would provide an opportunity for the authors to offer additional clarifications if needed.

Cheers,

AC

---

### Decision · Program_Chairs · 2023-09-21

**Decision:**

Accept (poster)

**Comment:**

All reviewers found the studied problem to be important, the proposed method to be effective, and the results promising. The rebuttal successfully addressed reviewer comments and all reviewers recommend acceptance. The authors are encouraged to improve the final paper version by following reviewer recommendations.